# Estimation of the geometric measure of entanglement with Wehrl moments through artificial neural networks

Jérôme Denis, François Damanet and John Martin⋆

Institut de Physique Nucléaire, Atomique et de Spectroscopie,
CESAM, University of Liège, B-4000 Liège, Belgium

⋆ jmartin@uliege.be

## Abstract

In recent years, artificial neural networks (ANNs) have become an increasingly popular tool for studying problems in quantum theory, and in particular entanglement theory. In this work, we analyse to what extent ANNs can accurately predict the geometric measure of entanglement of symmetric multiqubit states using only a limited number of Wehrl moments (moments of the Husimi function of the state) as input, which represents partial information about the state. We consider both pure and mixed quantum states. We compare the results we obtain by training ANNs with the informed use of convergence acceleration methods. We find that even some of the most powerful convergence acceleration algorithms do not compete with ANNs when given the same input data, provided that enough data is available to train these ANNs. We also provide an experimental protocol for measuring Wehrl moments, which is state-independent. More generally, this work opens up perspectives for the estimation of entanglement measures and other SU(2)-invariant quantities, such as the Wehrl entropy, in a way that is more accessible in experiments than by means of full state tomography.



## Contents

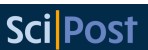

# 1  Introduction

Entanglement is at the heart of quantum physics and constitutes a crucial resource for most quantum technologies [1]. Detecting and estimating the entanglement of a system is usually a challenging task, both theoretically and experimentally, and the development of theoretical methods and experimental protocols are essential in this context. The detection of entanglement has already been explored around specific symmetric multiqubit states [2, 3] or using criteria based on collective measurements [3] or PPT mixtures [4] that are able to detect certain classes of entanglement. In this work, we propose a method for estimating the entanglement of *symmetric* multiqubit states, but we make no a priori assumptions about the form of the states or their entanglement.

More precisely, we tackle the problem of estimating entanglement via the use of artificial neural networks (ANNs). Over the past few years, deep learning methods have gained momentum in quantum physics [5, 6]. In the context of quantum state tomography, they have been used to reconstruct density matrices from measurement results [7,8] and to find an optimal measurement basis [9]. In quantum optics, artificial neural networks have been trained to detect multimode Wigner negativity [10]. Deep reinforcement learning and recurrent neural networks have also been exploited for quantum information theory purposes, such as quantum state preparation [11] and quantum error-correction [12,13].

In the context of entanglement theory, ANNs have been used to quantify the amount of entanglement in multipartite quantum systems [14, 15] and to classify the entanglement in pure states [16] and mixed states [17]. In [14], the authors trained complex-valued ANNs to predict the geometric measure of entanglement (GME) of symmetric states. To do so, they reformulated the GME computational problem as the search for the best rank-one tensor approximation of complex tensors, for which they used ANNs. Other authors have used deep learning methods to compute the concurrence and mutual information from an incomplete tomography of mixed qubit states [15]. In quantum many-body physics, convolutional neural networks were employed to compute e.g. the entanglement entropy from the variance on the number of particles in an electron chain [18].

More specifically, the general question posed in this work, which is along these lines, is: *To what extent is it possible to estimate the geometric measure of entanglement of symmetric multiqubit states using only partial information in the form of some of their Wehrl moments?* Wehrl moments are the moments of the Husimi $Q$ function of a state [19]. They have been used to define measures of non-classicality, chaoticity or entropy of quantum states [19–21], and have some relevance in various contexts, such as for the characterization of quantum phase transitions [21, 22]. Importantly, Wehrl moments are experimentally accessible quantities, as we show in this work, from projection measurements of collective observables (see [24] for a full state tomography protocol). On the other hand, there is currently no protocol to determine the GME experimentally other than by full-state tomography, and its calculation, even for pure symmetric states, cannot generally be performed analytically and requires numerical optimisation. A good estimate of the GME on the basis of more readily available partial information than the full quantum state is therefore of theoretical and practical interest, and motivates our approach. In this work, assuming the knowledge of a few Wehrl moments of symmetric multiqubit states, we present and compare three different approaches to estimate their GME, one of which being an ANN that we found to be the most efficient. Note that similar but distinct issues to the one addressed in this work have recently been studied with respect to the detection and certification of entanglement from the Peres-Horodecki criterion based on the first moments of the partial transpose of a state [25, 26].

Our paper is organised as follows. In Sec. 2, we define the Husimi function, the Wehrl moments, the GME and their relations to each other for pure symmetric multiqubit states. In Sec. 3, we present how we generated the datasets of Wehrl moments used throughout this work. In Sec. 4, we introduce the three different approaches to estimate the GMEs of the dataset: i) a first one based on the two highest known successive Wehrl moments, ii) a second one based on a convergence acceleration algorithm applied on the sequence of the known Wehrl moments and iii) a third one based on a trained ANN. In Sec. 5, we compare and analyse our results. In Sec. 6, we consider the more complex case of mixed states. In Sec. 7, we propose a protocol for the experimental determination of Wehrl moments based on the measurement of a set of collective observables, the number of which varies only quadratically with the number of qubits. In Sec. 8, we conclude and present perspectives of our work. Finally, this manuscript ends with a series of technical appendices, one of which presents a semi-definite program for the calculation of the GME of mixed multiqubit symmetric states (Appendix E).

# 2 Wehrl moments and geometric measure of entanglement

In this section, we define multiqubit symmetric states, the Husimi function and the associated Wehrl moments, the GME, and present how these quantities are related to each other.

## 2.1 Multiqubit symmetric states

A multiqubit state is said to be symmetric if it is invariant under any permutation of the qubits. Let $|\psi\rangle$ be an $N$-qubit symmetric state. We can always write this state in terms of $N$ single-qubit normalized states $|\epsilon_i\rangle$ as

$$|\psi\rangle = \mathcal{N}_{|\psi\rangle} \sum_{\sigma \in S_N} |\epsilon_{\sigma(1)}\rangle \otimes |\epsilon_{\sigma(2)}\rangle \otimes \cdots \otimes |\epsilon_{\sigma(N)}\rangle, \tag{1}$$

where $\mathcal{N}_{|\psi\rangle}$ is a normalization constant and $S_N$ is the symmetric group on $N$ elements. Since a one-qubit state, up to a phase factor, can be represented by a point on the Bloch sphere, any symmetric multi-qubit state can be represented geometrically by a constellation of $N$ points, each associated with one of the $|\epsilon_i\rangle$, on the same sphere [27]. In the following, we will refer to these points as the Majorana points of $|\psi\rangle$.

Alternatively, a symmetric state of $N$ qubits can be expanded in the symmetric Dicke states basis as

$$|\psi\rangle = \sum_{k=0}^{N} d_k |D_N^{(k)}\rangle, \tag{2}$$

where the symmetric Dicke states $|D_N^{(k)}\rangle$ are given by Eq. (1) with $|\epsilon_i\rangle = |1\rangle$ for $i = 1, \ldots, k$ and $|\epsilon_i\rangle = |0\rangle$ for $i = k+1, \ldots, N$. The states $|D_N^{(k)}\rangle$ can be thought as angular momentum eigenstates once we introduce the collective spin operators associated with the $N$-qubit system, $J_k = \frac{1}{2} \sum_{i=1}^{N} \sigma_k^{(i)}$ with $k = x, y, z$ and $\sigma_k^{(i)}$ the Pauli operators $\sigma_k$ for qubit $i$. It then holds that $J^2 |D_N^{(k)}\rangle = j(j+1)|D_N^{(k)}\rangle$ and $J_z |D_N^{(k)}\rangle = m|D_N^{(k)}\rangle$ with $j = N/2$ and $m = N/2 - k$.

## 2.2 Husimi function and Wehrl moments

### 2.2.1 Husimi function

For a spin $j$, the Husimi function of an arbitrary state $|\psi_j\rangle$ is defined as $Q_{|\psi_j\rangle}(\Omega) = |\langle \psi_j | \Omega \rangle|^2$, where $|\Omega\rangle$ is a spin-coherent state with $\Omega$ specifying a point on the unit sphere of $\mathbb{R}^3$ [28]. The Husimi function $Q_{|\psi_j\rangle}(\Omega)$ is an infinitely differentiable function on the sphere $S^2$. In what follows, we will mainly use the notation $Q_{|\psi_j\rangle}(\theta, \varphi)$ where $\theta \in [0, \pi]$ and $\varphi \in [0, 2\pi[$ are the polar and azimuthal angles associated to a point on the unit sphere. The Husimi function is normalized according to [28]

$$\frac{1}{4\pi} \int_{S^2} Q_{|\psi_j\rangle}(\Omega) \, d\Omega = \frac{1}{2j+1}. \tag{3}$$

For multiqubit symmetric states, the Husimi function $Q_{|\psi\rangle}(\theta, \varphi)$ of an $N$-qubit state $|\psi\rangle$ is similarly defined as the overlap squared of $|\psi\rangle$ with a symmetric separable pure state $|\epsilon\rangle^{\otimes N}$ where $\Omega = (\theta, \varphi)$ are the coordinates of the point on the Bloch sphere associated with the *single-qubit* state $|\epsilon\rangle \equiv |\theta, \varphi\rangle$. The Husimi function of any state $|\psi\rangle$ is normalized according to (3) with $|\psi_j\rangle \to |\psi\rangle$ and $2j \to N$. Using Eq. (1), we can expand it as

$$Q_{|\psi\rangle}(\theta, \varphi) = (N! \mathcal{N}_{|\psi\rangle})^2 \, |\langle \epsilon_1 | \theta, \varphi\rangle|^2 \, |\langle \epsilon_2 | \theta, \varphi\rangle|^2 \cdots |\langle \epsilon_N | \theta, \varphi\rangle|^2. \tag{4}$$

The Husimi function of three different symmetric states of $N = 8$ qubits are shown in Figure 1.

### 2.2.2 Wehrl moments – explicit expressions

The Wehrl moment $W_{|\psi\rangle}^{(q)}$ of integer order $q$ is the SU(2) invariant defined as

$$W_{|\psi\rangle}^{(q)} = \frac{1}{4\pi} \int_{S^2} \left( Q_{|\psi\rangle}(\Omega) \right)^q d\Omega. \tag{5}$$

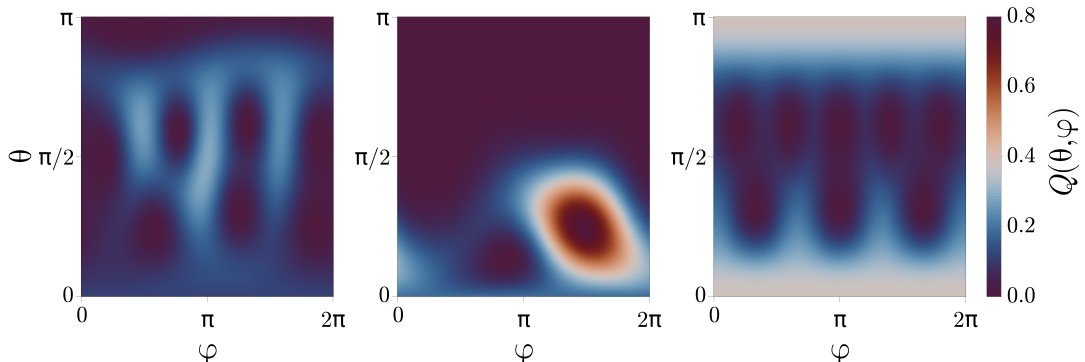

Figure 1: Husimi $Q$ function of symmetric 8-qubit states taken from the three different data subsets introduced in Sec. 3. From left to right (subsets 1 to 3), the GME is 0.717, 0.211 and 0.620 respectively. The more uniform the Husimi function, the higher the GME.

A tight upper bound for Wehrl moments of order $q > 1$ that is valid for any state is given by [28]

$$W_{|\psi\rangle}^{(q)} \leqslant \frac{1}{Nq+1} \, , \tag{6}$$

where the equality holds only for coherent states [29].

An explicit expression for the Wehrl moments of symmetric multiqubit states in terms of expansion coefficients $d_k$ in the Dicke states basis has been given by Gnutzmann and Zyczkowski [19], and reads in our notations

$$W_{|\psi\rangle}^{(q)} = \sum_{m=0}^{qN} \frac{1}{qN+1} \binom{qN}{m}^{-1} \left| \sum_{i_1,\dots,i_q} \prod_{k=1}^{q} \sqrt{\binom{N}{i_k}} d_{i_k} \right|^2 \, , \tag{7}$$

where the inner sum goes from 0 to $N$ for each $i_k$ with the restriction $\sum_{k=1}^{q} i_k = m$. This relation is exact and allows us to calculate the Wehrl moments when we know the expansion (2) of a symmetric state. In Appendix A, we give an alternative expression of Wehrl moments in terms of permanents of Gram matrices of constituent states $\{|\epsilon_i\rangle\}_{i=1}^{N}$, see Eq. (A.11). The latter expression is more appropriate when a symmetric state is known in the form of Eq. (1) rather than Eq. (2).

## 2.3 Geometric measure of entanglement

The geometric measure of entanglement (GME) of an $N$-qubit pure state $|\psi\rangle$, denoted by $E_G(|\psi\rangle)$, quantifies how far $|\psi\rangle$ is from the set of separable states. Just as the Wehrl moments, it is an SU(2) invariant quantity, defined as [30]

$$E_G(|\psi\rangle) = 1 - \max_{\{|\phi_i\rangle\}_{i=1}^{N}} |\langle \phi_1 \otimes \phi_2 \cdots \otimes \phi_N |\psi\rangle|^2 \, , \tag{8}$$

where the maximization is performed over the $N$ single-qubit states $|\phi_i\rangle$. The GME is always smaller than 1 and is equal to 0 only when $|\psi\rangle$ is separable. In the case of symmetric states, the maximization appearing in Eq. (8) can be replaced by the simpler maximization where all single qubit states $|\phi_i\rangle$ are identical, i.e. $|\phi_i\rangle = |\epsilon\rangle$ for $i = 1, \dots, N$ [31]. We are thus left with the problem of finding the maximum of the Husimi function of $|\psi\rangle$ on the sphere $S^2$, that is

$$\max_{|\epsilon\rangle} |\langle \epsilon \otimes \epsilon \cdots \otimes \epsilon |\psi\rangle|^2 = \max_{\substack{\theta \in [0,\pi] \\ \phi \in [0,2\pi[}} Q_{|\psi\rangle}(\theta, \varphi) \, . \tag{9}$$

The GME is zero for all product states and non-zero for all entangled states. An (not tight) upper bound on the GME of $N$-qubit symmetric states is given by [32]

$$E_G(|\psi\rangle) \leqslant 1 - \frac{1}{N+1}. \tag{10}$$

## 2.4 Bounds on GME from Wehrl moments

For any integers $q > p > 1$ and any state $|\psi\rangle$, it holds that

$$\max_{\theta,\phi} Q_{|\psi\rangle} \geqslant \frac{W_{|\psi\rangle}^{(q+1)}}{W_{|\psi\rangle}^{(q)}} \geqslant \frac{W_{|\psi\rangle}^{(p+1)}}{W_{|\psi\rangle}^{(p)}}. \tag{11}$$

This is a consequence of the integral Hölder's inequality [33],

$$\|f\,g\|_1 \leqslant \|f\|_r \|g\|_m, \tag{12}$$

where $\|f\|_r = \left(\int_X |f|^r d\mu\right)^{\frac{1}{r}}$, $r, m \in [1,\infty]$ with $1/r + 1/m = 1$, and $f$ and $g$ are functions defined on $X$. By taking $f = Q_{|\psi\rangle}$, $g = Q_{|\psi\rangle}^q$, $X = S^2$, $d\mu = d\Omega/4\pi$, $r = \infty$ and $m = 1$, we readily get Eq. (11) by noting that $\|f\|_\infty = \max_X f$ where $\|\cdot\|_\infty$ denotes the spectral norm. Equation (11) provides us with a chain of better and better upper bounds for the GME as $q$ and $p$ increase. In fact, defining the sequence (for integer $q > 1$)

$$S_{|\psi\rangle}(q) = \frac{W_{|\psi\rangle}^{(q)}}{W_{|\psi\rangle}^{(q-1)}}, \tag{13}$$

we have that

$$E_G(|\psi\rangle) \leqslant 1 - S_{|\psi\rangle}(q), \quad \forall\, q > 1, \tag{14}$$

and

$$E_G(|\psi\rangle) = 1 - \max_{\theta,\phi} Q_{|\psi\rangle} = 1 - \left\|Q_{|\psi\rangle}\right\|_\infty = 1 - \lim_{q\to\infty} S_{|\psi\rangle}(q). \tag{15}$$

Equation (15) shows that the geometric measure of entanglement $E_G$ can be extracted from the limit of the sequence $S_{|\psi\rangle}(q)$ of ratios of successive Wehrl moments.

The Wehrl moments admit in some cases simple analytical expressions. For instance, for symmetric Dicke states, they are given by [19]

$$W_{|D_N^{(k)}\rangle}^{(q)} = \frac{\binom{N}{k}^q}{(Nq+1)\binom{Nq}{kq}}. \tag{16}$$

This then leads to

$$1 - E_G(|D_N^{(k)}\rangle) = \lim_{q\to\infty} S_{|D_N^{(k)}\rangle}(q) = \begin{cases} \binom{N}{k}\left(\frac{k}{N}\right)^k\left(\frac{N-k}{N}\right)^{N-k}, & 0 < k < N-1, \\ 1, & k = 0 \vee k = N, \end{cases} \tag{17}$$

in agreement with known results for the geometric entanglement of Dicke states [30]. It is also instructive to analyze how the sequence $S_{|D_N^{(k)}\rangle}(q)$ converges to its limit. From Eq. (16) for $0 < k < N-1$, we find that the sequence $S_{|D_N^{(k)}\rangle}(q)$ is monotonously decreasing and converges asymptotically to its limit as

$$S_{|D_N^{(k)}\rangle}(q) = S_{|D_N^{(k)}\rangle}(\infty)\left[1 - \frac{1}{2q} + \frac{\frac{2}{k-N} - \frac{2}{k} + \frac{26}{N} - 3}{24q^2} + \mathcal{O}\left(\frac{1}{q^3}\right)\right]. \tag{18}$$

For separable states ($k = 0$ or $k = N$), we have [19]

$$W^{(q)}_{\text{coh}} = \frac{2j+1}{2qj+1} \quad \Rightarrow \quad \lim_{q\to\infty} S_{\text{coh}}(q) = 1 \,, \tag{19}$$

and

$$S_{\text{coh}}(q) = S_{\text{coh}}(\infty)\left[1 - \frac{1}{q} + \frac{1}{Nq^2} + \mathcal{O}\left(\frac{1}{q^3}\right)\right]. \tag{20}$$

In both cases, the dominant correction scales as $1/q$.

The asymptotic scaling as $1/q$ of the dominant correction of $S_{|\psi\rangle}(q)$ is actually a general feature of the sequence valid for *any* state $|\psi\rangle$. Indeed, the asymptotic scaling of the Wehrl moments (5) can be calculated using Laplace's method (see Appendix B for a detailed derivation) and reads

$$W^{(q)}_{|\psi\rangle} = c_{|\psi\rangle} \frac{\left\|Q_{|\psi\rangle}\right\|^q_\infty}{q}(1 + o(1)), \tag{21}$$

where $c_{|\psi\rangle}$ is a constant independent of $q$ and $o(\cdot)$ the little-o notation.[1] From the definition (13) and properties of the little-o and Big-o, we get

$$S_{|\psi\rangle}(q) = \left\|Q_{|\psi\rangle}\right\|_\infty \left(1 + \mathcal{O}\left(\frac{1}{q}\right)\right), \quad \forall\, |\psi\rangle \,. \tag{22}$$

In Section 4.2, we show how to generalize this analysis and how to take advantage of the knowledge of the asymptotic behavior of the sequence $S_{|\psi\rangle}(q)$ to estimate its limit from a finite number of terms.

## 3 Datasets and performance metrics

As our objective is to compare different methods to determine the best estimate of the GME of a state from its first few Wehrl moments, we need a set of representative pure multiqubit states on which to test these methods and calculate some metrics to compare their respective performances (see Sec. 4). This section aims to explain how we generated these representative multiqubit states and what our performance measures are.

### 3.1 Generation of datasets

In order to obtain a dataset with the most distributed GME values, we generate three different subsets of states. Subset 1 is made of symmetric states with randomly and uniformly distributed Majorana points on the Bloch sphere. Subset 2 is made of random states for which *degenerated* Majorana points are uniformly distributed on the Bloch sphere, with random degeneracy tuples drawn uniformly from all partitions of $N$. Finally, the subset 3 is made of superpositions of $|\text{GHZ}\rangle = (|D_N^{(0)}\rangle + |D_N^{(N)}\rangle)/\sqrt{2}$ and Dicke states, i.e.

$$|\psi(\alpha, k)\rangle = \mathcal{N}\left[\alpha\,|\text{GHZ}\rangle + (1 - \alpha)\,|D_N^{(k)}\rangle\right], \tag{23}$$

with random real number $\alpha \in [0, 1]$ and random integer $k$ between 0 and $N$. For each number of qubits $N$, 20000 states are randomly drawn for each subset. All these states are then divided into two equally sized sets: one for training the ANN and the other for testing the three different methods in the estimation of the GME. The Wehrl moments up to $q_{\text{max}} = 8$ and $E_G$ are computed for all states.

---

[1]A function $f(q)$ is "little-o" of $g(q)$, i.e., $f(q) = o(g(q))$, as $q \to \infty$ if $\lim_{q\to\infty} f(q)/g(q) = 0$. A function $f(q)$ is "Big-o" of $g(q)$, i.e., $f(q) = \mathcal{O}(g(q))$, as $q \to \infty$ if $\exists M : |f(q)| \leqslant Mg(q)$ in some neighborhood of $\infty$.

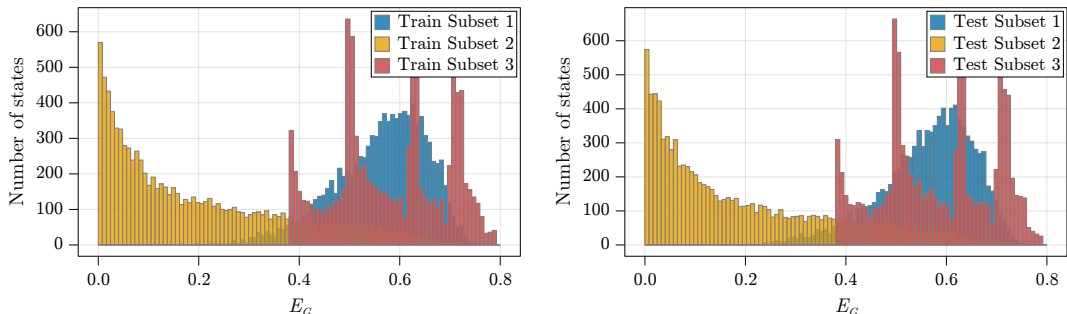

Figure 2: Frequency distributions of GME of the training set (left) and test set (right) for $N = 8$ qubits, where the three subsets of states are represented by different colors. The number of states in the data sets is large enough to generate a similar GME distribution for the training and test sets. For $N = 8$, the maximal GME is $E_G \approx 0.816$ [34], while Eq. (10) gives the upper bound $E_G(|\psi\rangle) \leqslant 8/9 \approx 0.889$.

Figure 2 shows the GME probability distribution of training states (left) and test states (right) for $N = 8$. We find that these three subsets have very different entanglement distributions and are therefore a good set of training and test data. In particular, subset 2 (yellow histograms) is mostly made up of weakly entangled states, while subset 3 (red histograms) contains a significant proportion of very highly entangled states.

## 3.2  Performance metrics

In order to compare the different methods to estimate the GME, such as convergence acceleration processes and ANNs, we first define the relative difference between the predicted GME and the actual GME as

$$\delta_i = \frac{E_G(|\psi_i\rangle) - E_G^{\text{pred}}(|\psi_i\rangle)}{E_G(|\psi_i\rangle)}, \tag{24}$$

where $E_G^{\text{pred}}(|\psi_i\rangle)$ stands for the predicted GME of state $|\psi_i\rangle$ of the test dataset. Then, we define the mean absolute relative difference, hereafter called *mean relative error* (MRE),

$$\Delta = \frac{1}{M} \sum_{i=1}^{M} |\delta_i|, \tag{25}$$

where we sum over all states of the test dataset of size $M = 30\,000$. As the distribution of the absolute relative difference $|\delta_i|$ is not Gaussian, the standard deviation is not a good estimate for error bars. Instead, we calculate a low error bar and a high error bar so as to include 68.2% of the $|\delta_i|$ distribution in the error bar and have 15.9% of the distribution below (above) the low (high) error bar, as would be the case for an interval of one standard deviation centred around the mean for a Gaussian distribution.

## 4  Estimation of the geometric measure of entanglement

In this section, we estimate the GME of the states of the test dataset presented previously based on the knowledge of their Wehrl moments from $q = 1, \ldots, q_{\text{max}}$, expecting a better estimate of the GME as $q_{\text{max}}$ increases. We use and compare three different methods: i) a crude one based on the ratio of the two highest known Wehrl moments, ii) a second one based on a convergence acceleration algorithm applied on the set of known Wehrl moments and iii), a third one based

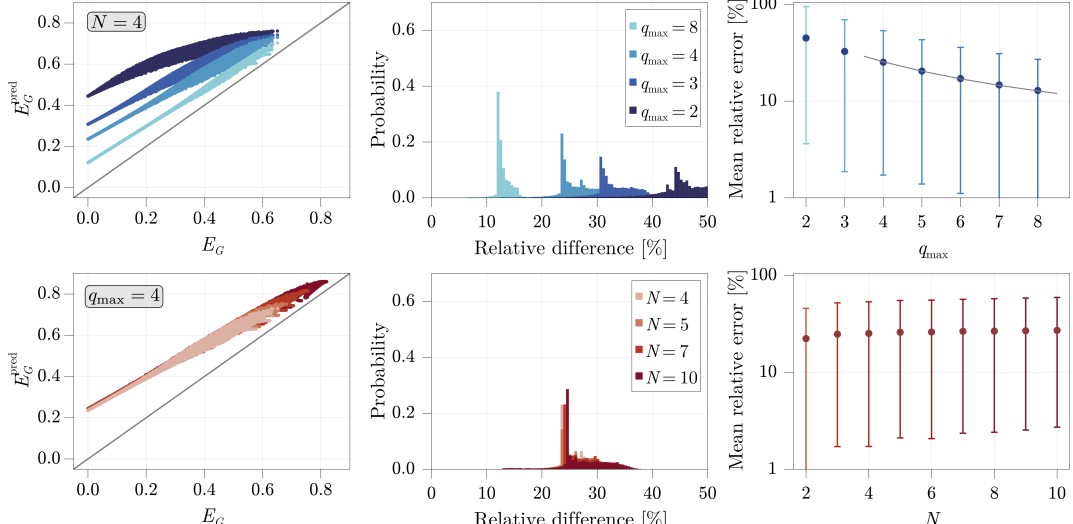

Figure 3: Predictions of $E_G$ based directly on Wehrl moment ratios $S_{|\psi\rangle}(q_{max}) = W_{|\psi\rangle}^{(q_{max})}/W_{|\psi\rangle}^{(q_{max}-1)}$ for $N = 4$ and different maximal orders $q_{max}$ (top) and for $q_{max} = 4$ and different number of qubits $N$ (bottom). Left panels: Predicted value versus actual value of GME for all states of the test dataset. Middle panels: Probability to predict the GME with a certain relative difference. The bins size is 0.5%. Right panels: Mean relative error (25) as a function of $q_{max}$ and $N$. The grey solid line in the top right panel shows a fit of equation $\Delta(q_{max}) = A/q_{max}$ with $A \approx 102$, which is the expected behaviour at large $q_{max}$ according to Eq. (22).

on a trained ANN. We are particularly interested in the performance of the different methods as a function of the highest considered order $q_{max}$ and of the number of qubits $N$.

## 4.1 Wehrl moments ratios

As the ratios of successive Wehrl moments (13) converge to the maximum of the Husimi function when $q \to \infty$ [see Eq. (15)], a first estimate of the GME of the test states based on these ratios is given by

$$E_G^{pred}(|\psi\rangle) = 1 - S_{|\psi\rangle}(q_{max}).$$ (26)

The predictive power of (26) is illustrated in Fig. 3 for different maximal orders $q_{max}$ and number of qubits $N$. As expected from the inequality (14), we observe that the estimate (26) is always larger than the actual value of the GME (left panels), which results in a positive relative difference (middle panels). As $q_{max}$ increases, the estimate becomes better and better, with a decrease in mean relative error (MRE) as a function of $q_{max}$ (top right panel). However, even with $q_{max} = 8$, the MRE remains above 10%. The MRE increases slightly with $N$ before stabilising quickly, as shown in the bottom right panel.

## 4.2 Convergence acceleration algorithms

Convergence acceleration algorithms consist in transforming a sequence into another sequence that converges faster to its limit, by taking as inputs only the first terms of the original sequence. Different algorithms exist in the literature and differ from each other depending on how the terms of the initial sequence are combined together to generate the new sequence. We focus here on the use of the recursive $E$-algorithm [35], which is among the algorithms we tested the one that showed the best performance. Our goal, by applying it on the sequence

$S_{|\psi\rangle}(q)$ [Eq. (13)], is to obtain a better estimate of its limit $S_{|\psi\rangle}(\infty)$, and thus of the GME of the states through Eq. (15).

The recursive $E$-algorithm makes it possible to accelerate sequences $f(q)$ with asymptotic expansions of the general form

$$f(q) = f(\infty)\big[1 + \lambda_1 g_1(q) + \lambda_2 g_2(q) + \lambda_3 g_3(q) + \dots\big], \tag{27}$$

where $g_i(q)$ are known (or postulated) scaling functions ordered such that

$$\lim_{q\to\infty} \frac{g_{i+1}(q)}{g_i(q)} = 0, \quad \forall\, i, \tag{28}$$

i.e., so that $g_1(q)$ corresponds to the dominant asymptotic scaling of the sequence $f(q)$, and with arbitrary (and potentially unknown) coefficients $\lambda_i$. According to the recursive $E$-algorithm, a better estimate of the limit $f(\infty)$ can be obtained by computing via recurrence the quantities

$$E_k^{(q)} = \frac{E_{k-1}^{(q)} g_{k-1,k}^{(q+1)} - E_{k-1}^{(q+1)} g_{k-1,k}^{(q)}}{g_{k-1,k}^{(q+1)} - g_{k-1,k}^{(q)}}, \tag{29}$$

taking $E_0^{(q)} = f(q)$ as the initial conditions and the coefficients

$$g_{k,i}^{(q)} = \frac{g_{k-1,i}^{(q)} g_{k-1,k}^{(q+1)} - g_{k-1,i}^{(q+1)} g_{k-1,k}^{(q)}}{g_{k-1,k}^{(q+1)} - g_{k-1,k}^{(q)}}, \qquad g_{0,i}^{(q)} = g_i(q), \qquad \forall\, k \in \mathbb{N}_0,\, i \geqslant k+1. \tag{30}$$

A quick inspection shows that $E_k^{(q)}$ is a function of the set $\{f(q), f(q+1), \dots, f(q+k)\}$. In practice, increasing the order $k$ of the algorithm generally provides a better estimate $E_k^{(q)}$ of the limit $f(\infty)$ of the initial sequence $f(q)$, but requires knowing and combining more terms of the sequence.

The recursive $E$-algorithm is particularly suited for the acceleration of the sequence $S_{|\psi\rangle}(q)$ for which we have an idea of the form of the scaling functions $g_i(q)$ defined in Eq. (27). Indeed, motivated by the general asymptotic behaviour of $S_{|\psi\rangle}(q)$ given by Eq. (22) and the two particular cases (18) and (20) studied in Sec. 2.4, we consider here the following ansatz:

$$S_{|\psi\rangle}(q) = S_{|\psi\rangle}(\infty)\left(1 + \frac{\lambda_1}{q} + \frac{\lambda_2}{q^2} + \frac{\lambda_3}{q^3} + \dots\right), \tag{31}$$

i.e., the general expansion (27) with $g_i(q) = q^{-i}$.

In Fig. 3, we showed the GMEs of the states of the test dataset via the crude estimate $S_{|\psi\rangle}(q_{\max})$. In order to have a fair comparison, we estimate here the GMEs of these states with $E_{q_{\max}-2}^{(2)}$, which exploits all the first terms of the sequence $S_{|\psi\rangle}(q)$ up to $q = q_{\max}$, i.e., $\{S_{|\psi\rangle}(q) : q = 2, \dots, q_{\max}\}$. Figure 4 shows the results for different $N$ and $q_{\max}$. As expected, the estimates of the GME is better than the crude estimate $S_{|\psi\rangle}(q_{\max})$ with the convergence acceleration algorithm, especially for low $q_{\max}$. In particular, the skewness of the distributions of predicted GMEs compared to actual GMEs is much less pronounced. For $N = 4$, we can see in the top right panel that the MRE is already reduced to only about 10% for $q_{\max} = 3$ [one order of magnitude lower than for the estimate $1 - S_{|\psi\rangle}(q_{\max})$]. For larger $q_{\max}$, we find a behaviour compatible with an exponential decrease of the MRE.

Note that we also compared the results of the $E$-algorithm to the ones obtained via the implementation of the $\theta$-algorithm, a popular convergence acceleration algorithm which has the advantage to not require the knowledge of the asymptotic scaling of the accelerated sequence, but we did not find better performance (data not shown).

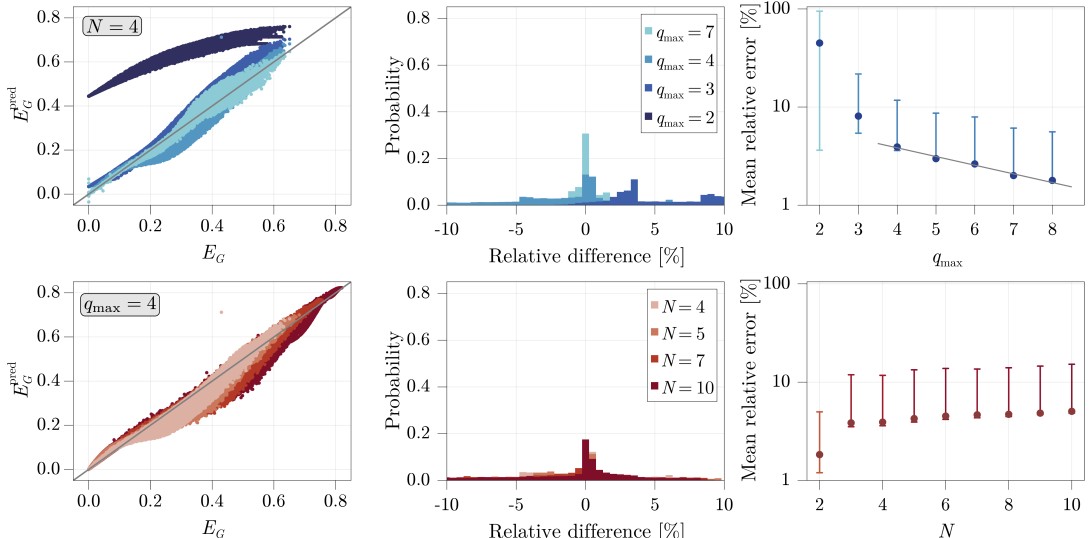

Figure 4: Same representation and parameters as in Fig. 3, but with predictions based on the recursive $E$-algorithm. The grey solid line in the top right panel shows a decreasing exponential fit of equation $\Delta(q_{max}) \approx 8.667 \exp(-0.204\, q_{max})$.

### 4.3 Artificial neural networks

One of the great advantages of ANNs is their predictive power in non-linear regression problems. Here we are interested in the ability of an ANN to predict the GME based on a few Wehrl moments. Basically, a neural network is a set of layers (see e.g. Fig. 5), indexed by $l$, containing a given number $N_l$ of nodes, indexed by $i$, each containing a real value $y_i^{(l)}$. Each node is linked to the nodes in the nearest layers by weights $w_{ij}^{(l)}$. The values contained in the first layer are the input data $y_i^{(0)}$. In this work, $y_i^{(0)} \equiv S_{|\psi\rangle}^{(i+1)}$. Each value of these nodes is propagated to the nodes of the next layer by multiplying it by the weight connecting the two nodes.

Therefore, the values of the nodes in the first layer are as follows

$$y_i^{(1)} = \sum_{j=1}^{N_0} w_{ij} y_j^{(0)}. \tag{32}$$

To increase the capability and predictive power of the network, a bias $b_i^{(l)}$ can be added to each node and, in order to obtain a non-linear regression, a non-linear function $f$ can be applied to each value in a given layer. Thus, the general form of the values contained in layer $l$ is

$$y_i^{(l)} = f\left(\sum_{j=1}^{N_{l-1}} w_{ij} y_j^{(l-1)} + b_i^{(l)}\right). \tag{33}$$

By feeding the nodes of one layer with the values of the previous layer, the input data flows through the network and finally the last layer contains the value of the regression, in this case an estimate of the GME. Initially, the weights and biases are chosen randomly. In the training process, the neural network updates them using the gradient descent algorithm in order to minimise a given loss function that compares the expected result and the value of the last layer.

For the learning process, we take a batch size of 500 and, for each $q_{max} \in [2,8]$ and $N \in [2,10]$, we train the ANN in a supervised manner for 5000 epochs with the ADAM optimizer. Our loss function is the squared difference averaged over the batch. Remarkably,

even after 5000 epochs, no overfitting is observed (see Fig. 11 and the additional discussion in Appendix C).

We now want to train artificial neural networks (ANNs) so that when we feed them with the finite sequence

$$\left\{ S_{|\psi\rangle}(q) : q = 2, \ldots, q_{\max} \right\},$$

for some state $|\psi\rangle$, they output an estimate for $E_G(|\psi\rangle) = 1 - \lim_{q\to\infty} S_{|\psi\rangle}(q)$, as schematically represented in Fig. 5. To be able to compare the trainings based on different $q_{\max}$ and $N$, we choose to always use the same network architecture

$$(q_{\max} - 1, 512, \text{ReLU}, 256, \text{ReLU}, 128, \text{ReLU}, 64, \text{ReLU}, 32, 1), \tag{34}$$

where ReLu is the nonlinear Rectified Linear Unit as used in deep learning [36].

We show in Fig. 6 the results of the different trainings applied to the test dataset. We find that ANNs give quite reliable predictions already for $q_{\max} = 3$ with a MRE at 1%, one order of magnitude less than with the convergence acceleration. More surprisingly, even on the basis of the first non-trivial Wehrl moment $W_{|\psi\rangle}^{(2)}$, ANNs give a good estimate for weakly and strongly entangled states. When we take into account more Wehrl moments, the ANNs are able to predict the GME more accurately. For a fixed number $q_{\max} = 4$ of Wehrl moments (see Appendix C for $q_{\max} = 8$), we find that the MRE increases as we increase the number of qubits but eventually saturates. We believe that for a higher number of qubits, there is a greater spectrum of states with the same first Wehrl moments but different GMEs. This would imply that the input to the ANN is not sufficient to distinguish between these different states and would explain the observed increase in error. We also observe that at $q_{\max} = 4$, the MRE saturates at about 1% for $N \gtrsim 5$. This result is quite remarkable as it shows that with ANNs the MRE seems to scale very favourably with $N$.

## 5 Discussion of the main results

We will now summarise our main results. We show in Fig. 7 the mean relative error for the different methods investigated in Sec. 4, for a wide range of maximum orders $q_{\max}$ and number of qubits $N$. The relative performance of the different methods of obtaining estimates for the GME are clearly evident. We consistently find that the MRE on the GME is lowest for the ANNs, then for the convergence acceleration algorithm and finally for the Wehrl moment ratios. The differences in performance are quite large, with ANNs outperforming the other

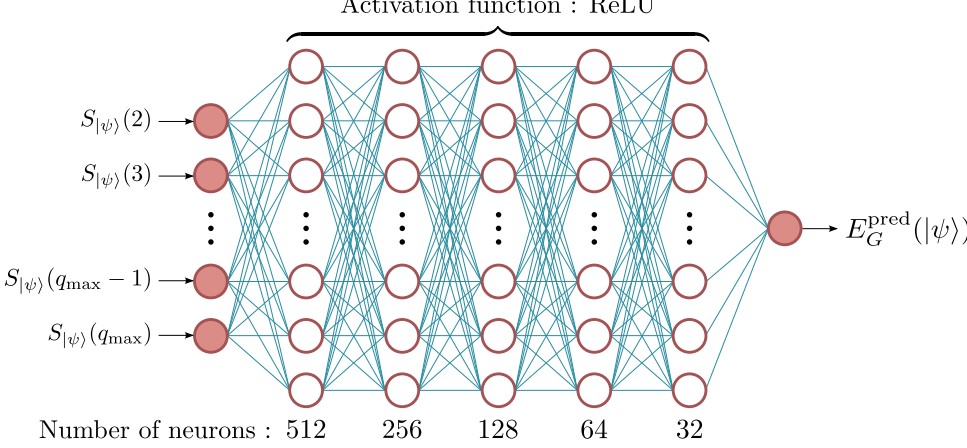

Figure 5: Representation of the ANN architecture used in this work.

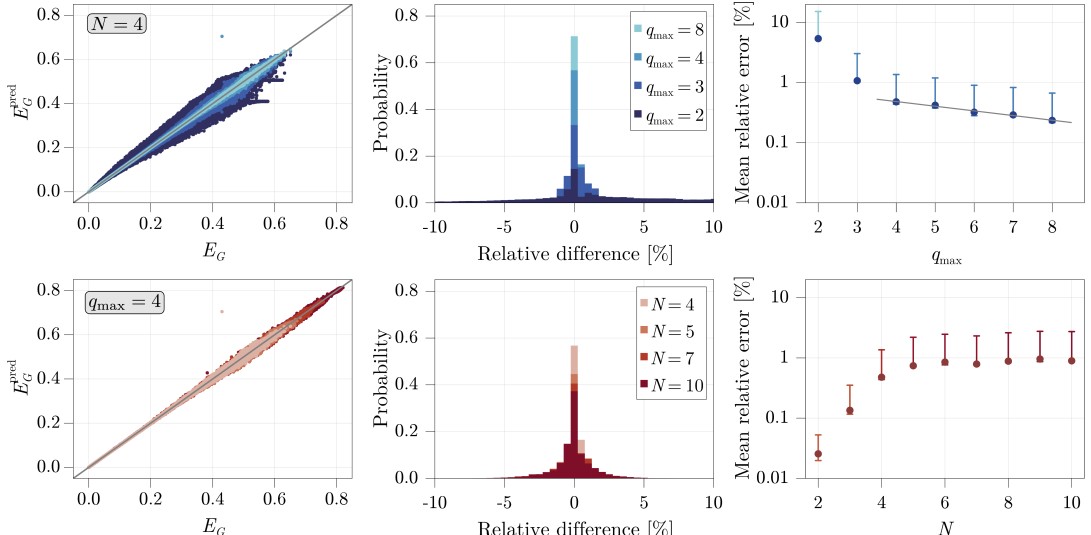

Figure 6: Same representation and parameters as in Fig. 3, but with predictions based on trained ANNs. The grey solid line in the top right panel shows a decreasing exponential fit of equation $\Delta(q_{max}) \approx 0.989 \exp(-0.179\, q_{max})$.

methods by at least an order of magnitude. For the methods based on ANNs and convergence acceleration algorithms, the MRE decreases very rapidly from $q_{max} = 2$ to $q_{max} = 4$. Then, the MRE decreases exponentially at roughly the same rate for both methods. For $q_{max} = 4$, the MRE obtained with ANNs seems to quickly saturate to about 1% for large number of qubits ($N \gtrsim 6$, see right panel). We have also tested the ANN on a set of pure states that have been dynamically generated from spin squeezing. This set is characterised by a GME distribution that differs strongly from those used to train the ANN (see appendix C for more details). In this case, we find that the ANN also works very well with similar performance, demonstrating its great flexibility upon variations of input data. Furthermore, we show in Appendix D that an ANN trained on noisy Wehrl moments is still able to predict the GME quite accurately.

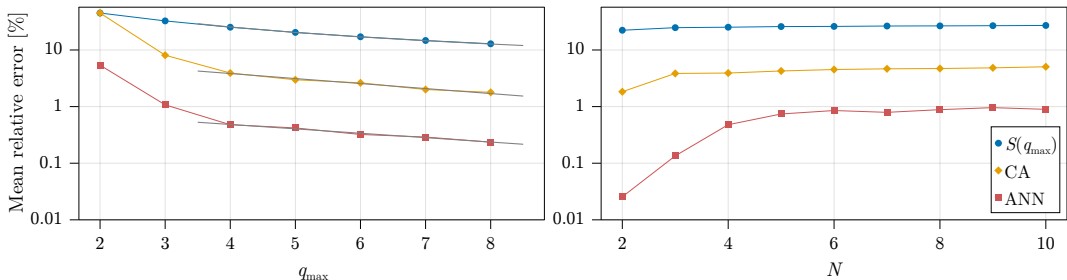

Figure 7: Comparison of the mean relative error (MRE) on the GME obtained with the bare Wehrl moment ratios (blue dots), with the recursive $E$-algorithm for convergence acceleration (yellow diamonds) and with ANNs (red squares). Left panel: MRE as a function of $q_{max}$ for $N = 4$. Right panel: MRE as a function of $N$ for $q_{max} = 4$.

# 6 Extension to mixed states

Under experimental conditions, the quantum state of a system is never perfectly pure due to the interaction of the system with its environment, resulting for example in depolarisation. It is therefore important to address the case of mixed states as well. Although the relationship (15) between Wehrl moments and GME is only valid for pure states, Wehrl moments can nevertheless provide valuable information about mixed states and potentially also about their entanglement. Therefore, it is still interesting to try to train ANNs to predict the GME of mixed states on the basis of their Wehrl moments. Note that for a mixed state $\rho$, the Wehrl moments are defined as in Eq. (5) with the Husimi function now given by $Q_\rho(\Omega) = \langle \Omega | \rho | \Omega \rangle$.

## 6.1 GME for mixed states

The geometric measure of entanglement of a mixed state $\rho$ is defined based on the convex roof construction

$$E_G(\rho) = \min_{\{p_i, |\psi_i\rangle\}} \sum_i p_i E_G(|\psi_i\rangle), \tag{35}$$

where the minimum is taken over all pure state decompositions $\{p_i, |\psi_i\rangle\}$ of $\rho$. In [37], it was shown that this definition is equivalent to another definition based on the distance of $\rho$ to the convex set $\mathcal{S}$ of separable mixed states,

$$E_G(\rho) = 1 - \max_{\sigma_{\text{sep}} \in \mathcal{S}} F\left(\rho, \sigma_{\text{sep}}\right), \tag{36}$$

where

$$F(\rho, \sigma) = \text{Tr}\left(\sqrt{\sqrt{\rho}\,\sigma\,\sqrt{\rho}}\right)^2, \tag{37}$$

is Uhlmann's fidelity between any two mixed states $\rho$ and $\sigma$. The form (36) allows us to compute the GME of mixed states using a semidefinite program, as we explain in Appendix E (see also [38]).

## 6.2 Results for depolarized states

For training the network, we generated a set of 1000 depolarised mixed states for each $N \in \{2, 3, 4\}$ and reduced the batchsize to 50. The mixed states were obtained by drawing pure random states $|\psi\rangle$ according to the Haar measure and mixing them with the maximally mixed state $\rho_0 = \mathbb{1}/(N+1)$ as follows

$$\rho = (1-k)|\psi\rangle\langle\psi| + k\rho_0, \tag{38}$$

where $k \in [0, 1]$ is a parameter quantifying the degree of depolarisation. The results on the test data are represented in Fig. 8 for $k = 0.05$ by yellow diamonds. We see that for depolarised states, the MRE is around 0.1% or even below for $N \in \{2, 3\}$ and below 1% for $N = 4$ for $q_{\text{max}} \geq 4$. For comparison, we also show the lower MRE obtained for pure random states (see Section 4.3) by blue dots. The data displayed in Fig. 8 shows that Wehrl moments remain useful quantities for predicting entanglement of mixed states in multiqubit systems. It is interesting to note that even for highly mixed states of the form (38), ANNs are still able to predict with high accuracy the GME. This is shown in Fig. 9, where we consider higher degrees of depolarisation $k$. Counter-intuitively, we find that the predictions on the GME improve as $k$ increases (see middle and right panel). This is probably due to the specific class of mixed states we have considered and the fact that the range of GME values that the ANN has to account for decreases with $k$ (see left panel). It does, however, show that for a typical decoherence model such as depolarization, Wehrl moments still contain essential information for predicting the GME even for highly mixed states.

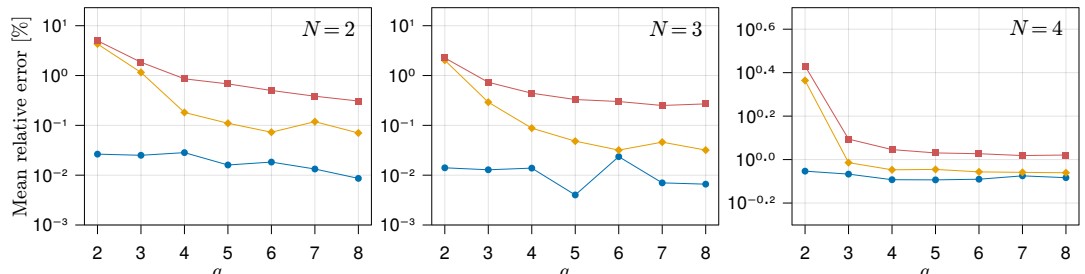

Figure 8: Results of the training of the ANNs on mixed states of the form (38) (yellow diamonds) and (39) (red squares) for $k = 0.05$. The blue dots represent the MRE for the predictions of the ANNs trained in Section 4.3 and applied to the pure states used in the equations (38) and (39) to generate the mixed states forming the test dataset.

We also trained ANNs on 1000 mixed states obtained by drawing pure random states $|\psi\rangle$ and mixed random states $\rho$ and mixing them as follows

$$\rho = (1-k)|\psi\rangle\langle\psi| + k\rho \, . \tag{39}$$

The results on the test data are represented in Fig. 8 by red squares. This time, the error is systematically higher than that obtained for the depolarised states (38), but it remains at an acceptable level for $k = 0.05$ and $q_{max} \geqslant 4$.

# 7 Measurement of Wehrl moments

## 7.1 Protocol for measuring Wehrl moments

In this section, we propose a simple protocol based on spherical $t$-designs that allows the experimental determination of Wehrl moments of various orders from the same set of measurement outcomes of Stern-Gerlach experiments. A spherical $t$-design is a set of $n_t$ points on the unit sphere, located at angles $\Omega_k = (\theta_k, \varphi_k)$ with $k \in \{1, \ldots, n_t\}$, such that [39, 40]

$$\frac{1}{4\pi} \int_{\mathcal{S}^2} P(\Omega) \, d\Omega = \frac{1}{n_t} \sum_{k=1}^{n_t} P(\Omega_k) \, , \tag{40}$$

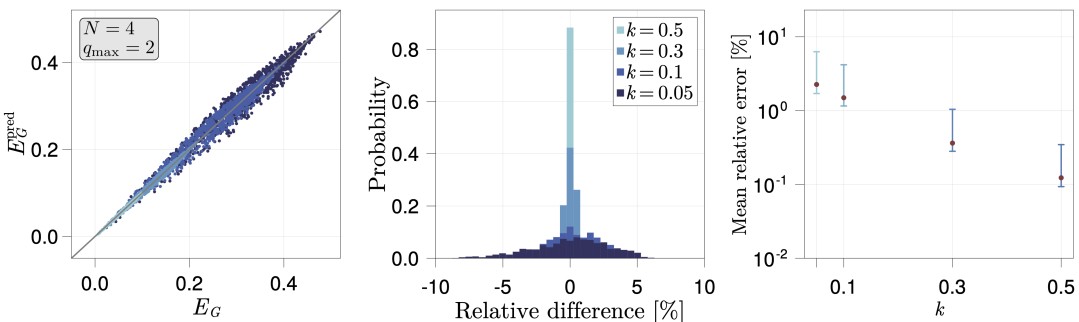

Figure 9: Results obtained by training ANNs on mixed states of the form (38) for different degrees of depolarisation $k \in \{0.05, 0.1, 0.3, 0.5\}$. Here, $N = 4$ and $q_{max} = 2$.

for *all* trigonometric polynomials $P$ of degree at most $t$. Taking $P(\Omega) = \left(Q_\rho(\Omega)\right)^q$, and assuming for the moment that $t$ is sufficiently large, we obtain by combining Eqs. (5) and (40)

$$W_\rho^{(q)} = \frac{1}{n_t} \sum_{k=1}^{n_t} \left(Q_\rho\left(\Omega_k\right)\right)^q, \tag{41}$$

from which we can conclude that it is sufficient to measure the Husimi function in a finite number $n_t$ of directions to determine the Wehrl moments. The Husimi function at $\Omega$ can be rewritten as

$$\begin{aligned}
Q_\rho(\Omega) &\equiv |\langle\Omega|\rho|\Omega\rangle|^2 = |\langle D_N^{(0)}|R(\Omega)^\dagger\rho R(\Omega)|D_N^{(0)}\rangle|^2 \\
&= |\langle D_N^{(0)}|\rho_\Omega|D_N^{(0)}\rangle|^2 \\
&= p_0^\Omega,
\end{aligned}$$

where $R(\Omega)$ is the non-entangling rotation operator which maps the separable Dicke state $|D_N^{(0)}\rangle$ to the product state $|\Omega\rangle$, $\rho_\Omega = R(\Omega)^\dagger \rho R(\Omega)$ is the rotated state and $p_0^\Omega$ is the probability that the system in state $\rho_\Omega$ is found in state $|D_N^{(0)}\rangle$. The latter probability can be measured from a Stern-Gerlach experiment giving access to $\left\{p_k^\Omega = |\langle D_N^{(k)}|\rho_\Omega|D_N^{(k)}\rangle|^2 : k = 0, \ldots, N\right\}$ or, in the case of an atomic system, by driving a dipole transition to an auxiliary energy level and then observing the resonance fluorescence to obtain $p_0^\Omega$ [41]. This protocol involving measurements of the Husimi function by determining the probability of a multi-qubit state being in different pure separable states *using rotations* is a fairly common technique, see e.g. [24, 42, 43], and can be applied for single spin systems, collections of two-level systems and even for light polarization. In fact, it has already been routinely implemented in several experiments [44–46], e.g. using half-wave plates and polarising beam splitters in the case of multiphoton polarization states [46].

The advantage of our protocol, which consists of measuring the Husimi function in a finite number of directions and extracting the Wehrl moments, is that it is totally independent of the state under consideration. Indeed, the Husimi function of any $N$-qubit symmetric state is a polynomial function of degree $N$. By choosing $t = Nq$, all Wehrl moments can be extracted exactly, up to order $q$, irrespective of the state $\rho$. As regards spherical designs, it has been shown numerically that $n_t \approx t^2/2$ [47], so to extract the Wehrl moments up to order $q$, we should measure the Husimi function in $\approx (Nq)^2/2$ points. This quadratic scaling with $N$ is clearly more favourable than the cubic scaling of full state tomography for multiqubit symmetric states [48, 49]. Note also that our protocol is not necessarily optimal and that there might be clever ways of using the full set of probabilities $\{p_k^\Omega\}$ obtained in Stern-Gerlach experiments (instead of only $p_0^\Omega$) to find a better approximation of the Wehrl moments.

## 7.2 Results from approximate Wehrl moments

Since ANNs are, to a certain extent, intrinsically robust to noise, it is not necessary to have perfect determination of the Wehrl moments in order to obtain good estimates of the GME (see Appendix D for more details). This suggests the possibility of using spherical designs of order $t$ less than $Nq$ to obtain approximate Wehrl moments up to order $q$ via

$$W_\rho^{(q)} \approx \frac{1}{n_t} \sum_{k=1}^{n_t} \left(Q_\rho\left(\Omega_k\right)\right)^q. \tag{42}$$

Equation (42) approximates all Wehrl moments with $q > t/N$ from the same set of Husimi function values. Therefore, as long as this improves the prediction of the ANN, we can give it approximate Wehrl moments of increasing order.

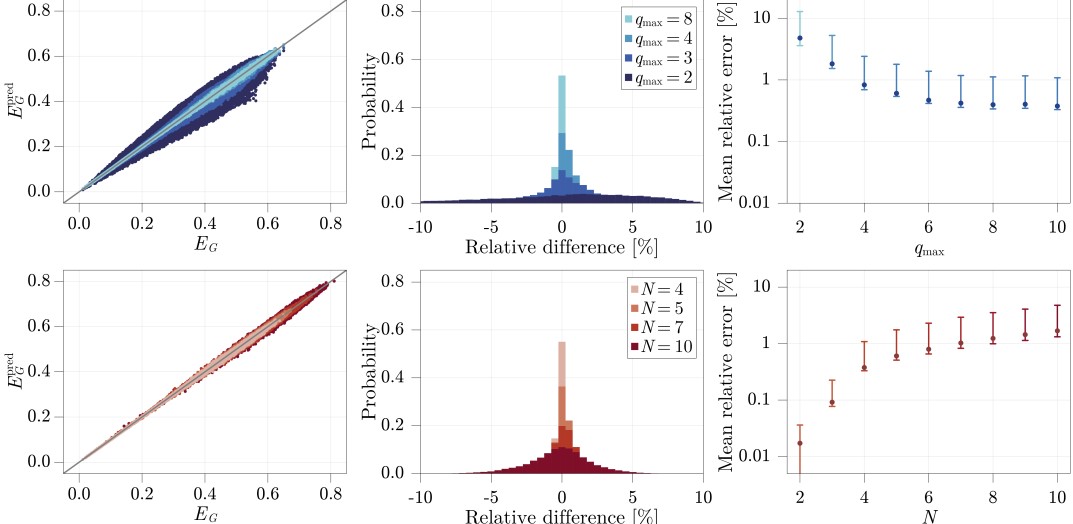

Figure 10: Same representation and parameters as in Fig. 3, but with predictions based on Wehrl moments obtained from Eq. (42) with $\Omega_k$ the points defining a spherical $t$-design with $t = 13$ and $n_t = 94$. For the lower panels, we took $q_{max} = 10$ instead of $q_{max} = 4$.

We show in Fig. 10 the results of the training of ANNs based on the spherical $t$-design with $t = 13$ and the same test set of pure states as presented in Sec. 3. They show that the MRE can be brought down to a level of 1% with $t = 13$ even for a number of qubits up to 10. We chose this particular value of $t$ because the spherical design contains antipodal points and the Husimi function at two antipodal points can be measured by a single Stern-Gerlach experiment. The number of directions in which the Stern-Gerlach experiment must be performed can therefore be halved in this case (from $n_t = 94$ to 47).

## 8 Conclusion

In this work, we have studied how ANNs can be used to give an accurate estimate of the geometric measure of entanglement (GME) of pure and mixed symmetric multiqubit states based on their first Wehrl moments (moments of their Husimi function). We also used convergence acceleration methods to estimate the GME. More specifically, we implemented the algorithm $E$ informed by the asymptotic behaviour of the Wehrl moments which we determined analytically. We found that even this powerful convergence acceleration algorithm is outperformed by ANNs when fed with the same input data. We proposed an experimental protocol for measuring Wehrl moments that offers a gain over full state tomography and we showed that it can be coupled with ANNs to obtain a good estimate of the GME. This provides opportunities for the experimental estimation and certification of entanglement on the basis of a few Wehrl moments.

This work opens up several perspectives. First, while we have focused on the determination of GME, our approach could have been used to determine e.g. Wehrl entropy [50, 51], as both GME and such entropy are based on Wehrl moments, opening up characterizations of quantum chaos and phase transitions via ANNs. Secondly, it is known that determining the GME of a quantum state is a considerably more complex task for mixed states than for pure states. Nevertheless, as we have shown in Sec. 6, the GME of a depolarised state can still be predicted with high accuracy from its first Wehrl moments. Remarkably, we even found that

GME predictions improve as the state purity decreases, probably because the entanglement also decreases in this case. It would be of great interest to know for which other types of mixed states ANNs also give reliable estimates of the GME. In addition, our approach could be generalized to non-symmetric many-body quantum states where one is confronted with the exponential many-body wall, as it can be expected that ANNs will also be perform well in this context [52]. More generally, an approach similar to the one used in this work could be followed to estimate the maximum or minimum of a continuous (quasi)probability distribution other than the Husimi function from its first moments, such as the Wigner function to explore the non-classicality of quantum spin states.

## Acknowledgments

Most of the computations were done with the Julia programming language, in particular using the Flux.jl package [53] and the Convex.jl package [54] with the SCS optimizer [55]. The figures were produced with the package Makie [56].

**Author contributions**   JM conceived the presented idea, made the general theoretical developments and supervised the project. JD and FD carried out the developments and computations relating to ANNs and the acceleration of convergence, respectively. All authors discussed the results and their analysis at all stages of the work and contributed to the writing of the manuscript.

**Funding information**   Computational resources were provided by the Consortium des Équipe-ments de Calcul Intensif (CÉCI), funded by the Fonds de la Recherche Scientifique de Belgique (F.R.S.-FNRS) under Grant No. 2.5020.11. FD acknowledges the Belgian F.R.S.-FNRS for financial support during this work.

## A   Explicit expression of Wehrl moments in terms of $|\epsilon_i\rangle$

Suppose we are given a symmetric state in the form of Eq. (1), i.e. in terms of normalized single-qubit states $|\epsilon_i\rangle$ (hereinafter referred to as constituent states) as

$$|\psi\rangle = \mathcal{N}_{|\psi\rangle} \sum_{\sigma \in S_N} |\epsilon_{\sigma(1)}\rangle \otimes |\epsilon_{\sigma(2)}\rangle \otimes \cdots \otimes |\epsilon_{\sigma(N)}\rangle, \tag{A.1}$$

where the normalization constant $\mathcal{N}_{|\psi\rangle}$ is given by

$$\mathcal{N}_{|\psi\rangle}^{-2} = N! \sum_{\sigma \in S_N} \langle \epsilon_1 | \epsilon_{\sigma(1)} \rangle \ldots \langle \epsilon_N | \epsilon_{\sigma(N)} \rangle. \tag{A.2}$$

In this Appendix, we show how to obtain an expression for the Wehrl moments directly in terms of the $|\epsilon_i\rangle$. First, let $G_{|\psi\rangle}$ be the matrix of overlaps between the single-qubit states, that is,

$$G_{|\psi\rangle} = \begin{pmatrix} \langle \epsilon_1 | \epsilon_1 \rangle & \cdots & \langle \epsilon_1 | \epsilon_N \rangle \\ \vdots & \ddots & \vdots \\ \langle \epsilon_N | \epsilon_1 \rangle & \cdots & \langle \epsilon_N | \epsilon_N \rangle \end{pmatrix}. \tag{A.3}$$

The matrix (A.3) is nothing but the Gram matrix of the constituent states $\{|\epsilon_i\rangle\}_{i=1}^N$, which was also introduced in Ref. [57] in connection with the problem of determining the geometric measure of entanglement of symmetric states. Then the normalization constant can be expressed

as [58,59]

$$\mathcal{N}_{|\psi\rangle}^{-2} = N! \operatorname{per}(G_{|\psi\rangle}) \quad \Leftrightarrow \quad \mathcal{N}_{|\psi\rangle} = \frac{1}{\sqrt{N! \operatorname{per}(G_{|\psi\rangle})}} \, , \tag{A.4}$$

where $\operatorname{per}(A)$ denotes the permanent of the matrix $A$, defined as

$$\operatorname{per}(A) = \sum_{\sigma \in S_N} \prod_{i=1}^{N} A_{i\sigma(i)} \, . \tag{A.5}$$

After these preliminary developments, let us show how to obtain the desired explicit expression for the Wehrl moments. Some of our reasoning follows similar lines to those in Ref. [60]. We begin by noting that any integer power $q$ of the Husimi function (4) can be written as

$$Q_{|\psi\rangle}^{q}(\theta, \varphi) = (N! \mathcal{N}_{|\psi\rangle})^{2q} \underbrace{|\langle \epsilon_1|\theta, \varphi\rangle|^2 \cdots |\langle \epsilon_1|\theta, \varphi\rangle|^2}_{q \text{ times}} \cdots \underbrace{|\langle \epsilon_N|\theta, \varphi\rangle|^2 \cdots |\langle \epsilon_N|\theta, \varphi\rangle|^2}_{q \text{ times}}, \tag{A.6}$$

with $|\theta, \varphi\rangle$ the state of a qubit whose corresponding point on the Bloch sphere has coordinates $(\theta, \varphi)$. Based on Eq. (4), it is easy to see that, up to a multiplicative constant, Eq. (A.6) is the Husimi function of the $(Nq)$-qubit symmetric state $|\psi_q\rangle$ with the same constituent states $|\epsilon_i\rangle$ as $|\psi\rangle$ but each now appearing $q$ times (i.e. each state $|\epsilon_i\rangle$ is $q$-fold degenerated). Indeed, it holds that

$$Q_{|\psi_q\rangle}(\theta, \varphi) = ((Nq)! \mathcal{N}_{|\psi_q\rangle})^2 |\langle \epsilon_1|\theta, \varphi\rangle|^{2q} \cdots |\langle \epsilon_N|\theta, \varphi\rangle|^{2q} \, , \tag{A.7}$$

from which follows the relation

$$Q_{|\psi\rangle}^{q}(\theta, \varphi) = \frac{(N! \mathcal{N}_{|\psi\rangle})^{2q}}{((Nq)! \mathcal{N}_{|\psi_q\rangle})^2} Q_{|\psi_q\rangle}(\theta, \varphi) \, . \tag{A.8}$$

Then, since the Husimi function $Q_{|\psi_q\rangle}(\theta, \varphi)$ obeys the normalization condition

$$\frac{1}{4\pi} \int Q_{|\psi_q\rangle}(\Omega) \, d\Omega = \frac{1}{Nq+1} \, , \tag{A.9}$$

we have

$$W_{|\psi\rangle}^{(q)} = \frac{1}{4\pi} \int Q_{|\psi\rangle}^{q}(\Omega) \, d\Omega = \frac{(N! \mathcal{N}_{|\psi\rangle})^{2q}}{((Nq)! \mathcal{N}_{|\psi_q\rangle})^2} \frac{1}{Nq+1} \, , \tag{A.10}$$

or, finally, by using Eq. (A.4)

$$\boxed{W_{|\psi\rangle}^{(q)} = \frac{\operatorname{per}(G_{|\psi_q\rangle})}{(\operatorname{per}(G_{|\psi\rangle}))^q} \frac{(N!)^q}{(Nq+1)!} \, ,} \tag{A.11}$$

where $G_{|\psi\rangle}$ is the Gram matrix (A.3) and $G_{|\psi_q\rangle}$ is a Gram matrix made of $q \times q$ identical blocks $G_{|\psi\rangle}$ as follows

$$G_{|\psi_q\rangle} = \begin{pmatrix} G_{|\psi\rangle} & \cdots & G_{|\psi\rangle} \\ \vdots & \ddots & \vdots \\ G_{|\psi\rangle} & \cdots & G_{|\psi\rangle} \end{pmatrix} . \tag{A.12}$$

Equation (A.11) is our exact result for the Wehrl moments as a function of the constituent states $|\epsilon_i\rangle$ that appear in Eq. (1).

# B Asymptotic behaviour of the Wehrl moments

In this Appendix, we derive the asymptotic scaling of the Wehrl moments, Eq. (21), using Laplace's approximation for evaluating integrals, following [61].

First, let us rewrite without restriction the Wehrl moments (5) as

$$W_{|\psi\rangle}^{(q)} = \frac{1}{4\pi} \int_{S^2} e^{-qf_{|\psi\rangle}(\Omega)} d\Omega, \tag{B.1}$$

where $f_{|\psi\rangle}(\Omega) = -\ln\left(Q_{|\psi\rangle}(\Omega)\right)$ is a function $f : S^2 \to \mathbb{R}$. For large $q$, we expect the integrand to be non-negligible only around the minimum of $f_{|\psi\rangle}(\Omega)$ (the maximum of $Q_{|\psi\rangle}(\Omega)$). For simplicity, we consider here the generic case where the minimum is unique, which is however not the case for all states. The idea to obtain the asymptotic behavior of the Wehrl moments as $q \to \infty$ is to perform a series expansion of $f_{|\psi\rangle}(\Omega)$ around its minimum. For convenience, we expand instead the function $\tilde{f}_{|\psi\rangle}(\Omega) = [f_{|\psi\rangle}(\Omega) - f_{|\psi\rangle}(\Omega^*)]/f''_{|\psi\rangle}(\Omega^*)$ around $\Omega^*$, the value of $\Omega = (\theta, \varphi)$ minimizing $f_{|\psi\rangle}(\Omega)$, where $f''_{|\psi\rangle}(\Omega)$ the Hessian matrix of $f(\Omega)$. Since $\tilde{f}_{|\psi\rangle}(\Omega^*) = 0$ and $\tilde{f}''_{|\psi\rangle}(\Omega) = \mathbb{1}$ where $\mathbb{1}$ is the identity matrix, the expansion of $\tilde{f}_{|\psi\rangle}(\Omega)$ around $\Omega^*$ simply reads

$$\tilde{f}_{|\psi\rangle}(\Omega) = \frac{1}{2}||\Omega - \Omega^*||^2 + o\left(||\Omega - \Omega^*||^2\right) = \frac{1}{2}||\Omega - \Omega^*||^2(1 + o(1)), \tag{B.2}$$

where $|| \cdot ||$ is the standard Euclidian norm and $o(\cdot)$ the little-o notation.[1] The Wehrl moment (B.1) then reads

$$W_{|\psi\rangle}^{(q)} = \frac{1}{4\pi} e^{-qf_{|\psi\rangle}(\Omega^*)} \int_{S^2} e^{-qf''_{|\psi\rangle}(\Omega^*)\frac{||\Omega - \Omega^*||^2}{2}(1+o(1))} d\Omega. \tag{B.3}$$

By making a change of variable $\tilde{\Omega} = \sqrt{qf''_{|\psi\rangle}(\Omega)}(\Omega - \Omega^*)$ where $\sqrt{f''_{|\psi\rangle}(\Omega)}$ is the positive square root of the Hessian matrix $f''_{|\psi\rangle}(\Omega)$, the integral becomes

$$W_{|\psi\rangle}^{(q)} = \frac{e^{-qf_{|\psi\rangle}(\Omega^*)}}{4\pi q\sqrt{\det\left(f''_{|\psi\rangle}(\Omega^*)\right)}} \int_{\sqrt{qf''(\Omega^*)}(S^2 - \Omega^*)} e^{-\frac{||\Omega - \Omega^*||^2}{2}(1+o(1))} d\tilde{\Omega}, \tag{B.4}$$

where $\det(\cdot)$ is the determinant. For large $q$, the region of integration tends to $\mathbb{R}^2$, and the integral becomes a standard 2D Gaussian integral equal to $2\pi/(1 + o(1)) = 2\pi(1 + o(1))$. Hence, the asymptotic behavior of the Wehrl moments finally reads

$$W_{|\psi\rangle}^{(q)} = c_{|\psi\rangle} \frac{e^{-qf(\Omega^*)}}{q}(1 + o(1)) = c_{|\psi\rangle} \frac{\left\|Q_{|\psi\rangle}\right\|_\infty^q}{q}(1 + o(1)), \tag{B.5}$$

where

$$c_{|\psi\rangle} = \frac{1}{2\sqrt{\det\left(f''_{|\psi\rangle}(\Omega^*)\right)}}, \tag{B.6}$$

is a constant independent of $q$.

# C Additional information on ANNs

Figure 11 shows an example of the evolution of the loss function on the test dataset throughout the training of the ANN for different numbers of qubits and $q_{max} = 4$. We observe no overfitting, with the loss function decreasing even after a large number of epochs.

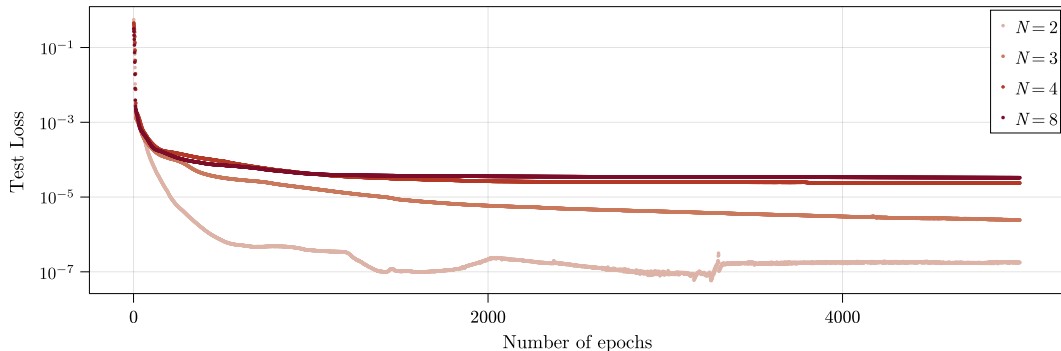

Figure 11: Loss function (averaged squared error, see Sec. 4.3) of the test dataset as a function of the number of training epochs for a maximal order $q_{max} = 4$ and different numbers of qubits $N$.

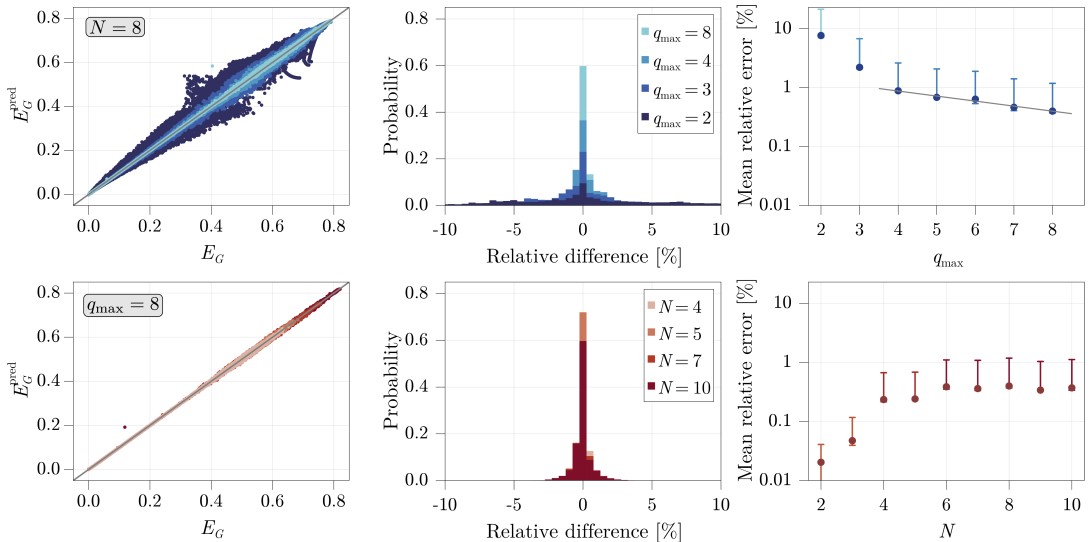

Figure 12: Same as Fig. 6 for $N = 8$ (top) and $q_{max} = 8$ (bottom). The grey solid line in the top right panel shows a decreasing exponential fit of equation $\Delta(q_{max}) \approx 1.919 \exp(-0.197\, q_{max})$.

Figure 12 shows the performance of the ANNs for a larger number of qubits and a larger maximal order than the results presented in the main text. For the top panels $N = 8$ and for the bottom panels $q_{max} = 8$. The same general observations as in the main text apply in this case, in particular the fact that the mean relative error is below 1% already for $q_{max} = 4$.

In order to further test the performance of ANNs, we generated another set of states resulting from the dynamical evolution corresponding to a spin squeezing. We calculated the time evolution of the initial coherent/product state $|D_N^{(0)}\rangle$ under the Hamiltonian

$$H = \chi_x J_x^2 + \chi_y J_y^2 + \chi_z J_z^2. \tag{C.1}$$

where $\chi_x, \chi_y, \chi_z$ are squeezing rates along the three spatial directions. At regular times, we sampled the state of the system and calculated its Wehrl moments and GME. After 500 time steps $\Delta t = 0.1$, we ended the evolution and started again from the same initial state. The $\chi_\alpha$ rates were chosen randomly between 0 and 1 at the beginning of each evolution. In this way, we generated 30 000 states on which we tested the previously trained ANNs. The results are presented in Fig. 13. We find that the ANNs still predict $E_G$ very well even though they have never handled this type of states before. This shows that the training set was sufficiently large

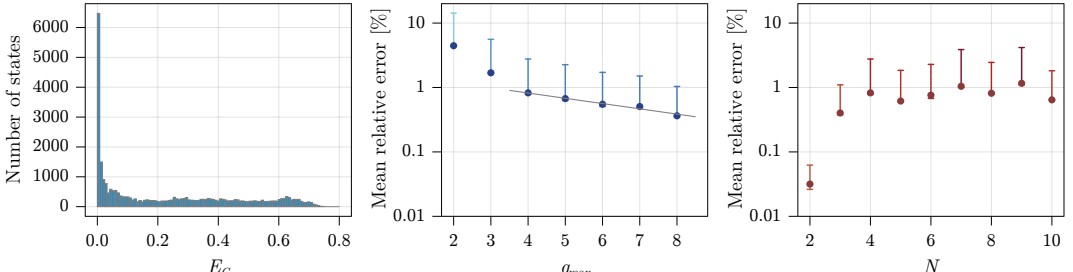

Figure 13: Left panel: Frequency distribution of GME of $30\,000$ squeezed states generated for $N = 8$ qubits. Middle and right panels: mean relative error on the estimate of the GME obtained from ANNs for $N = 4$ and $q_{\max} = 4$ respectively. The grey solid line shows a decreasing exponential fit of equation $\Delta(q_{\max}) \approx 1.745 \exp(-0.189\, q_{\max})$.

and representative to obtain ANNs capable of inferring beyond the states on which they have been trained.

## D Noisy Wehrl moments

In our previous developments, we used the exact value of the Wehrl moments for each multi-qubit state. However, the Wehrl moments may not be known exactly, e.g. because of noises that are inevitably present in an experiment or because they can only be calculated approximately. This provides an incentive to test ANNs with noisy inputs. As a first approach, we applied Gaussian noise to our inputs $S_{|\psi\rangle}(q)$ (from the same training and test data sets as before). More precisely, for each $q$, we first calculated the average value of the ratio of Wehrl moments over the whole data set, $\overline{S_{|\psi\rangle}(q)}$. Based on this value, we defined a normal distribution with a mean value of zero and a standard deviation given by

$$\sigma = \eta\, \overline{S_{|\psi\rangle}(q)}, \tag{D.1}$$

where $\eta$ is a real number that quantifies the magnitude of the noise. Then we applied noise, sampled from the normal distribution, to each Wehrl moment ratio and fed these noisy Wehrl moments to ANNs trained in two different ways: ANNs trained as before on noiseless Wehrl moments and ANNs trained directly on noisy Wehrl moments. The results are shown in Fig. 14 for $\eta = 0.01$. We find that the least satisfactory predictions are obtained from ANNs that have not been trained on noisy Wehrl moments (red squares). The explanation we see is that ANNs trained on noiseless Wehrl moments become excellent at predicting GME with such data but are unable to generalise on noisy data (a phenomenon similar to overfitting). However, ANNs trained on noisy Wehrl moments work much better and give a low mean relative error, around 1%, for $q_{\max} \geqslant 4$ (yellow diamonds). For a higher noise level, the MRE increases and is of the order of 2.6% for $\eta = 0.03$ with $q_{\max} = 4$ and $N = 4$.

# E  Semidefinite program for calculating the GME of mixed multi-qubit symmetric states

The computation of the geometric measure of entanglement (GME) of a mixed state $\rho$, which can be defined as [37]

$$E_G(\rho) = 1 - \max_{\sigma_{\text{sep}} \in \mathcal{S}} F\left(\rho, \sigma_{\text{sep}}\right), \tag{E.1}$$

where $F\left(\rho, \sigma_{\text{sep}}\right)$ is Uhlmann's fidelity between $\rho$ and $\sigma_{\text{sep}}$, involves an optimization on the convex set $\mathcal{S}$ of separable states. In Ref. [62], a method was derived to compute the maximum fidelity between a state $\rho$ and an arbitrary convex set of states $\mathcal{D}$ using semidefinite programming (SDP). This method is based on the equivalence between the problem of finding $\max_{\sigma \in \mathcal{D}} F(\rho, \sigma)$ and the SDP problem

$$\text{Find } \max_{\sigma \in \mathcal{D}, X} \left[\frac{1}{2}\text{Tr}(X) + \frac{1}{2}\text{Tr}\left(X^\dagger\right)\right], \quad \text{subject to } \begin{pmatrix} \rho & X \\ X^\dagger & \sigma \end{pmatrix} \geq 0, \tag{E.2}$$

where $X$ is a matrix with complex entries. Therefore, we only need a parametrization (even approximate) of the set of separable states $\mathcal{D} \equiv \mathcal{S}$ to be used in the SDP program (E.2) in order to be able to calculate the (approximate) value of the GME of mixed states. By Carathéodory's theorem (see e.g. [28]), we know that any separable symmetric state of $N$ qubits can be expressed as a convex sum of $(N+1)^2$ pure symmetric product states, that is

$$\sigma_{\text{sep}} = \sum_{i=1}^{(N+1)^2} p_i |\boldsymbol{\alpha}_i\rangle\langle\boldsymbol{\alpha}_i|, \tag{E.3}$$

with $|\boldsymbol{\alpha}_i\rangle \equiv |\alpha_i\rangle^{\otimes N}$ where $|\alpha_i\rangle$ are single-qubit states. But since the $|\boldsymbol{\alpha}_i\rangle$ in (E.3) are a priory not known, we can construct an ansatz for separable states by taking the convex combination of a large number $n_{\text{max}} \gg (N+1)^2$ of *fixed* pure product states $|\boldsymbol{\alpha}_i^{\text{rand}}\rangle$ drawn at random, i.e.

$$\sigma_{\text{sep}} = \sum_{i=1}^{n_{\text{max}}} p_i |\boldsymbol{\alpha}_i^{\text{rand}}\rangle\langle\boldsymbol{\alpha}_i^{\text{rand}}|, \tag{E.4}$$

where $p_i \geq 0$ and $\sum_i p_i = 1$. In our SDP problem, the $p_i$ and the entries of the $X$ matrix are then the variables to be optimised on. To perform the optimization, we used the Convex.jl package [54] written in Julia with the SCS optimizer [55]. We have verified that our SDP

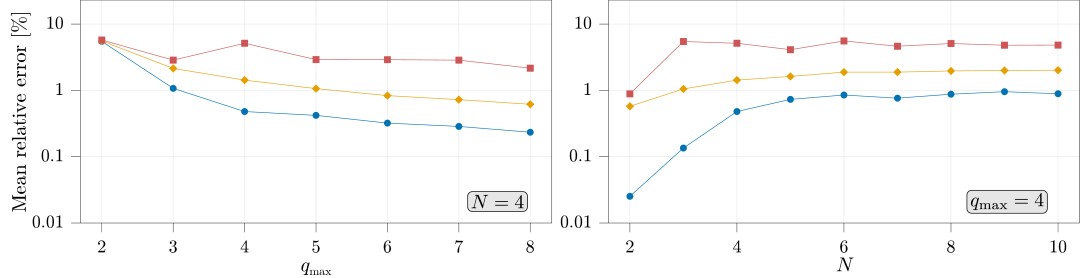

Figure 14: Mean relative error (MRE) on the GME obtained from ANNs fed with noisy input data. The red and yellow symbols give the MRE for ANNs trained respectively on noiseless and noisy Wehrl moments. For comparison, the blue dots give the MRE for ANNs trained and tested on noiseless Wehrl moments (see Fig. 6).

program works reliably for $2, 3$ and 4-qubit states with $n_{\max} = 1000$. In particular, we tested our SDP program on two-qubit isotropic states of the form

$$\rho_{\text{iso}} = \frac{1-p}{2}\mathbb{1} + \frac{3p-1}{2}|\text{GHZ}\rangle\langle\text{GHZ}|, \tag{E.5}$$

where $\mathbb{1}$ is the identity operator, $|\text{GHZ}\rangle$ is the 2-qubit GHZ state and $p \in [0.5 : 1]$. Their GME is given by [30]

$$E_G(\rho_{\text{iso}}) = 1 - \frac{1}{2}\left(\sqrt{p} + \sqrt{1-p}\right)^2, \tag{E.6}$$

a value that we found to an error of at most $10^{-5}$ for all $p \in [0.5 : 1]$.

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
