# Peer review of "Estimation of the geometric measure of entanglement with Wehrl Moments through Artificial Neural Networks"

_SciPost Physics, doi:SciPost Phys. 15, 208 (2023)_

## Round 1 · Referee Report · Anonymous (Referee 1) · 2022-9-12

Strengths

1-The paper provides a new and innovative approach to study entanglement, i.e., computing the geometric entanglement in symmetric states, which is a highly important set of states, using artificial neural networks.

2-It opens many interesting further research questions, some of which are also discussed in the conclusion.

Weaknesses

1-Although permutation symmetric states are interesting the authors only consider a very small class of states and pure states only. Experimentally realistic scenarios would however include mixed states as well.

2-It is not so clear what the limitations are of this approach, e.g., concerning its application to mixed symmetric states or even to states without any symmetries at all.

Report

In their article the authors study estimation techniques for the geometric measure of entanglement (GME) based on artificial neural networks. They prove strong bounds on the geometric measure in terms of Wehrl moments that in the limit resemble the GME. Wehrl moments have the advantage that they can be estimated in experiments without relying on full tomography. The authors then show how to estimate the GME from the first few moments only using convergence accelerated algorithms and ANNs. They find that the ANNs perform very well in this task even in cases that are very different from the training data (here states generated from spin squeezing).

The manuscript is very well written and concepts are thoroughly explained. I would accept the manuscript after small revisions (see below) based on the fact that it fulfils the acceptance criterion "Open a new pathway in an existing or a new research direction, with clear potential for multipronged follow-up work".

Requested changes

1- I think a general reader would greatly benefit from a more detailed introduction to ANNs. The idea of convergence acceleration algorithms is nicely explained for instance. I would ask the authors to provide at least a little background information in the general idea of ANNs.

2-Directly before section 5 the authors write "This [the saturation of RME with increasing number of qubits N] is probably due to the fact that the entanglement can take more complex forms for a higher number of qubits". It is definitely true that entanglement can take more complex forms if the particle numbers increases, but I cannot see how this is a probable cause for the saturation. Could the authors explain in more detail why they believe that this causes the RME to stabilize, or is there any, maybe numerical, evidence for this conclusion?

  • validity: top
  • significance: good
  • originality: high
  • clarity: top
  • formatting: excellent
  • grammar: excellent

Author:  John Martin  on 2023-01-12  [id 3232]

(in reply to Report 1 on 2022-09-12)
Category:
answer to question

The referee writes:

"1- I think a general reader would greatly benefit from a more detailed introduction to ANNs. The idea of convergence acceleration algorithms is nicely explained for instance. I would ask the authors to provide at least a little background information in the general idea of ANNs."

Our response

We thank the referee for this suggestion and have now added a description of the basic working principles of ANNs.

The referee writes:

"2-Directly before section 5 the authors write ”This [the saturation of RME with increasing number of qubits N] is probably due to the fact that the entanglement can take more complex forms for a higher number of qubits”. It is definitely true that entanglement can take more complex forms if the particle numbers increases, but I cannot see how this is a probable cause for the saturation. Could the authors explain in more detail why they believe that this causes the RME to stabilize, or is there any, maybe numerical, evidence for this conclusion?”

Our response

The referee is right to say that our explanation is not satisfactory as it stands. We believe that for a higher number of qubits, there is a greater spectrum of states with the same first Wehrl moments but different GMEs. This would imply that the input to the ANN is not sufficient to distinguish between these different states and would explain the observed increase in error. We have now modified the discussion in the manuscript accordingly.

The referee writes:

"1-Although permutation symmetric states are interesting the authors only consider a very small class of states and pure states only. Experimentally realistic scenarios would however include mixed states as well. 2-It is not so clear what the limitations are of this approach, e.g., concerning its application to mixed symmetric states or even to states without any symmetries at all.”

Our response

We appreciate the comments of the referee and have taken our study further by considering some mixed states as well. More specifically, we trained neural networks on Werhl moments of depolarized mixed states for small numbers of qubits (N = 2, 3, 4) in order to predict their GME that we calculated by Semi-Definite Programming. The results are again quite conclusive. They are presented in a new Section 6 and the method of calculating the GME is presented in the new Appendix E, as it may also be of interest to the reader.

---

## Round 1 · Referee Report · Anonymous (Referee 2) · 2022-10-17

Report

The authors consider the problem of estimating the entanglement of symmetric multiqubit pure (SMP) states from a given dataset. In particular, they assume to have access to the first few Wehrl moments (WMs) for a number of SMP states (where the latter are randomly sampled from three chosen distributions). Given the WMs up to a given order qmax, and assuming that they correspond to SMP states, the authors provide three methods to estimate their entanglement content as quantified by the geometric measure of entanglement (GME): 1) using the WM ratio of the largest available moments; 2) using an estimation of the limit of the sequence of the WM ratios (which the authors show to converge exactly to the GME); 3) using an artificial neural network (ANN) previously trained on a different dataset (of SMP states randomly sampled from the same distributions as for the test set). By a statistical analysis, the authors show that the latter method is the most powerful, being able to provide accurate estimates (i.e., within 1% of mean relative error) for qmax below 8 and for state composed of up to 10 qubits. From this analysis, the authors conclude that the combination of ANN and the knowledge of WMs is a powerful tool to estimate entanglement measures (and other SU(2) invariant quantities), based on the fact that WMs should be accessible experimentally with relative ease.

The manuscript is well written and organized. In Sec 2 and 3, the authors clearly made an effort to introduce the reader to the theoretical tools used in their analysis, which considerably ease the reading of the result sections. The results are scientifically sound and, in general, sufficient details are given for the numerical results to be reproducible. Also, the results presented are to the best of my knowledge original.

Despite the above, I have major concerns regarding the suitability of the manuscript for the present journal. Specifically, I think that the results presented do not meet the selective criteria for publication in SciPost Physics. In the conclusions the authors claim that their work open "several perspectives" for the experimental evaluation of entanglement measures in SMP states as well as in generic states, beyond SMP and possibly including mixed states. If such a claim were sufficiently supported by evidences I would have no reservation in recommending the manuscript for publication, as at least one of the SciPost criteria would be met ("Open a new pathway in an existing or a new research direction, with clear potential for multipronged follow-up work"). However, I do not see convincing arguments to support such a claim. In particular:

1) As said, the method is entirely based on WMs. The main motivation that the authors give to focus on a dataset composed of WMs is that they work under the assumption that it is relatively easy to measure them experimentally. In particular the authors claim that it is feasible in experiments to have access to the first few WMs for SMP states composed of several qubits. The evidence that is given to support such a claim is a set of references reported in the introduction: [17, 19, 22, 23]. However, these articles describe only theoretical proposals, and none of them demonstrate experimentally the measurement of WMs. In addition, to the best of my understanding, none of these articles addresses the multiqubit scenario studied in the present manuscript: Refs [19,22,23] concern continuous-variable systems, whereas Ref [17] focuses on a single qudit. Therefore, as said, I do not find in the manuscript sufficient support to the claim that WMs are easy to obtain experimentally, which is the practical motivation that underpins this analysis and, more importantly, the "perspectives" that it might open.

2) In the analysis reported, the WMs are assumed to be known exactly. In fact the analytical exact expression reported in Eq (16) is used in the manuscript. However, in a practical scenario, the dataset of WMs that should be used to feed the ANNs would only be known approximately [assuming that WMs can be somehow obtained, see point 1) above]. The impact of such uncertainty, which will most probably increase for increasing WM order qmax, is not considered in the present study. This should be addressed thoroughly to support the claim of usefulness of the method in a practical setting.

Besides the two major points above, there are some other relevant points that the authors should address to convincingly support their claims:

3) The authors speculate in the conclusions that their method opens new perspectives for generic states, possibly including mixed ones. While I understand that this is a speculation and that the authors do not intend to assert it with certainty, I still think that it would be useful if such a statement were supported by some preliminary evidence. It is in fact unclear to me if or how the ANN would generalize to the mixed case. The only hint at good generalization properties of the SMP-trained ANNs is given in the appendix C, where the ANN performances on a dynamically generated dataset are presented (see fig 10). These are still pure states though.

4) The authors claim that it is easier to measure WMs than perform full tomography (i.e., the approximate reconstruction of the Husimi function or any other representation of the entire state). Notwithstanding the fact that I agree with such a general statement for a generic state, I am not convinced that this is the case for the specific class of states at hand. In fact, SMPs are a very small subset of the set of pure multiqubit states. Therefore, it is reasonable to expect that full tomography will be significantly simpler to perform for SMPs than it is for the general case, via exploiting the symmetry of the states. Certainly, the scaling would not be exponential with the number of qubits. In other words, the approach via WMs might not be necessarily easier than full tomography in this case. The authors should give more arguments to support the contrary.

5) To put this work in context, it would be important to analyze more in details the connection with previous studies, reporting on possible advantages. I refer to the entanglement witness analysis reported previously for Dicke states and SMP states more in general, for example: - J. Opt. Soc. Am. B 24, 275 (2007) - New J. Phys. 11, 083002 (2009) - Phys. Rev. A 83, 040301(R) (2011) - Phys. Rev. A 88, 012305 (2013) - Phys. Rev. Applied 12, 044020 (2019)

In conclusion, I think that the present manuscript reports sufficient new results to be published in a specialized journal, however it does not meet the criteria for SciPost Physics. Therefore I cannot recommend it for publication.

  • validity: -
  • significance: -
  • originality: -
  • clarity: -
  • formatting: -
  • grammar: -

Author:  John Martin  on 2023-01-12  [id 3231]

(in reply to Report 2 on 2022-10-17)
Category:
answer to question
validation or rederivation

Measurement of WM

The referee writes:

"1) As said, the method is entirely based on WMs. The main motivation that the authors give to focus on a dataset composed of WMs is that they work under the assumption that it is relatively easy to measure them experimentally. In particular the authors claim that it is feasible in experiments to have access to the first few WMs for SMP states composed of several qubits. The evidence that is given to support such a claim is a set of references reported in the introduction: [17, 19, 22, 23]. However, these articles describe only theoretical proposals, and none of them demonstrate experimentally the measurement of WMs. In addition, to the best of my understanding, none of these articles addresses the multiqubit scenario studied in the present manuscript: Refs [19,22,23] concern continuous-variable systems, whereas Ref [17] focuses on a single qudit. Therefore, as said, I do not find in the manuscript sufficient support to the claim that WMs are easy to obtain experimentally, which is the practical motivation that underpins this analysis and, more importantly, the ”perspectives” that it might open."

Our response

Firstly, we have not been as firm in our statements as the referee claims when he/she writes that ”The main motivation that the authors give to focus on a dataset composed of WMs is that they work under the assumption that it is relatively easy to measure them experimentally.” In our introduction, we already stressed the purely theoretical relevance of our research question and emphasized the importance of Wehrl’s moments from a purely theoretical point of view. As we mentioned in our Introduction: ”They [Wehrl moments] have been used to define measures of non-classicality, chaoticity or entropy of quantum states [16, 18, 21], and have some relevance in various contexts, such as for the characterization of quantum phase transitions [20, 21].” The fact that Wehrl moments are experimentally accessible quantities is not strictly essential in our work, although it adds to its motivation. The main objective of our work is to determine theoretically whether the partial information about the state of a quantum system present in its first few Wehrl moments is enough to accurately estimate its degree of entanglement. This question is of relevance even in a theoretical scenario where the calculation of a few Wehrl moments would be more direct than the calculation of the full density matrix. Furthermore, we would like to stress that we have only written that Wehrl moments are experimentally accessible quantities that should be more easily accessible in experiments than the full state tomography, and no more than that. On the other hand, we agree that our claim that it is possible, in experiments, to access the first WMs for SMP states composed of several qubits was a bit hasty in view of the references we cited. We have rewritten part of the introduction to remove this ambiguity and make the purely theoretical interest of our work more obvious.

Advantage over full tomography

The referee writes:

"4) The authors claim that it is easier to measure WMs than perform full tomography (i.e., the approximate reconstruction of the Husimi function or any other representation of the entire state). Notwithstanding the fact that I agree with such a general statement for a generic state, I am not convinced that this is the case for the specific class of states at hand. In fact, SMPs are a very small subset of the set of pure multiqubit states. Therefore, it is reasonable to expect that full tomography will be significantly simpler to perform for SMPs than it is for the general case, via exploiting the symmetry of the states. Certainly, the scaling would not be exponential with the number of qubits. In other words, the approach via WMs might not be necessarily easier than full tomography in this case. The authors should give more arguments to support the contrary."

Our response

Indeed the full tomography of symmetric multiqubit states scales as \(\mathcal{O}[(N+1)^3]\) as was recently shown in M. Perlin et al. Phys. Rev. A 104, 062413 (2021). For a general mixed state, the Wehrl moment of order q = 2 takes the form

\begin{equation*} W_{\rho}^{(2)} = \sum_{L=0}^{N+1}\frac{(2j)!^2}{(2j-L)!(2j+L+1)!}\sum_{M=-L}^L|\rho_{LM}|^2 \end{equation*}

where \(\rho_{LM}\) are the state multipoles. Because \(|\rho_{00}|^2=\frac{1}{N+1}\), in order to determine the Wehrl moment of order 2 of a state, we only need to know the relative importance of consecutive multipoles \(\sum_{M=-(L+1)}^{L+1}|\rho_{L+1M}|^2/\sum_{M=-L}^L|\rho_{LM}|^2\) in numbers scaling as \(\mathcal{O}(N)\), which is of course much less than knowing all the state multipoles in numbers scaling as \(\mathcal{O}(N^2)\). Hence, even for symmetric states, this suggests there is an advantage of using the knowledge of a few WMs over the knowledge of the full state. We have added a paragraph in the main text to emphasize this.

Precision on WM

The referee writes:

"2) In the analysis reported, the WMs are assumed to be known exactly. In fact the analytical exact expression reported in Eq (16) is used in the manuscript. However, in a practical scenario, the dataset of WMs that should be used to feed the ANNs would only be known approximately [assuming that WMs can be somehow obtained, see point 1) above]. The impact of such uncertainty, which will most probably increase for increasing WM order qmax, is not considered in the present study. This should be addressed thoroughly to support the claim of usefulness of the method in a practical setting."

Our response

The uncertainties on the value of the Wehrl moments will depend on the specific approach to evaluate them theoretically or obtain them from experimental results. Here, we have only demonstrated a proof of concept of the use of neural networks for the estimation of entanglement based on Wehrl moments. The wide use of neural networks for non-linear regression in real-world problems is an indication that they are naturally robust to noise. Nevertheless, at the request of the referee, we have now tested our neural networks on noisy Wehrl moments, and have added a brief discussion in the main text and our conclusive results in an appendix of the revised manuscript.

Class of states

The referee writes:

"3) The authors speculate in the conclusions that their method opens new perspectives for generic states, possibly including mixed ones. While I understand that this is a speculation and that the authors do not intend to assert it with certainty, I still think that it would be useful if such a statement were supported by some preliminary evidence. It is in fact unclear to me if or how the ANN would generalize to the mixed case. The only hint at good generalization properties of the SMP-trained ANNs is given in the appendix C, where the ANN performances on a dynamically generated dataset are presented (see fig 10). These are still pure states though."

Our response

We appreciate the comments of the referees and have taken our study further by considering some mixed states as well. More specifically, we trained neural networks on Werhl moments of depolarized mixed states for small numbers of qubits (N = 2, 3, 4) in order to predict their GME that we calculated by Semi-Definite Programming. The results are again quite conclusive. They are presented in a new Section 6 and the method of calculating the GME is presented in the new Appendix E, as it may also be of interest to the reader.

Other minor points

The referee writes:

"5) To put this work in context, it would be important to analyze more in details the connection with previous studies, reporting on possible advantages. I refer to the entanglement witness analysis reported previously for Dicke states and SMP states more in general, for example: * J. Opt. Soc. Am. B 24, 275 (2007) * New J. Phys. 11, 083002 (2009) * Phys. Rev. A 83, 040301(R) (2011) * Phys. Rev. A 88, 012305 (2013) * Phys. Rev. Applied 12, 044020 (2019)"

Our response

We thank the referee for pointing out previous works on the entanglement of symmetric and permutation invariant states. We have briefly mentioned the links and differences with some of them in the introduction.

---

## Round 2 · Referee Report · Anonymous (Referee 1) · 2023-3-13

Report

In the revised manuscript the basic concepts behind ANNs are much more clear now. The other points that I raised in my previous report are also addressed satisfactorily. I appreciate that the authors discuss mixed states and the advantage over full tomography in the revised version. Therefore, I recommend the publication of this article.
  • validity: -
  • significance: -
  • originality: -
  • clarity: -
  • formatting: -
  • grammar: -

Author:  John Martin  on 2023-07-05  [id 3780]

(in reply to Report 1 on 2023-03-13)

We thank again the referee for recommending the publication of the article as it is. We hope that the elements we have just added will make the recommendation even stronger.

---

## Round 2 · Referee Report · Anonymous (Referee 2) · 2023-5-21

Report

As stated in my previous report, my primary critique concerned the lack of a convincing path towards experimental implementability for the method introduced in this work, as claimed by the authors. I also emphasized that if such a path were provided, I would recommend publication based on the high standard criteria of SciPost Physics ("Open a new pathway in an existing or a new research direction, with clear potential for multipronged follow-up work").

In response to my comments, the authors acknowledged that their previous claims about the experimental accessibility of Wehrl moment (WM) measurements were not supported. They stated in their reply that "we agree that our claim that it is possible, in experiments, to access the first WMs for SMP states composed of several qubits was a bit hasty in view of the references we cited."

Consequently, the authors revised some crucial sentences in their manuscript. For instance, they removed the sentence "Importantly, Wehrl moments are experimentally accessible quantities" from the introduction. Furthermore, in contrast to the initial version, the revised abstract no longer mentions the experimental accessibility of their approach.

Based on this, the authors clarified in their reply that the primary motivation for their work is theoretical in nature: "the main objective of our work is to determine theoretically [...]; "This question is of relevance even in a theoretical scenario where [...]."

I appreciate the authors' effort to clarify that their results are, as they stand, of interest only from a theoretical viewpoint. However, this fact can only reinforce my previous critique. In other words, notwithstanding the originality of their findings, I cannot find convincing evidences that their work can open a new research pathway or meet the high standard of this journal.

To be clear, I do not imply that a theoretical work cannot meet the high standard of SciPost Physics. However, firstly, the original version of this work suggested opening a new research pathway based on the experimental accessibility of the new method -- a claim that has now been withdrawn by the authors themselves, as said. Secondly, as a purely theoretical work, I do not find the method introduced here to be innovative and relevant enough:

  • The method has only a limited advantage compared to full tomography for the considered states. The advantage seems to be only quadratic [see for example the reply to point 4) of my previous report].

  • There is little to no quantitative comparison with at least some of the vast literature of theoretical methods proposed to estimate (or witness) entanglement for this class of states [see point 5) of my previous report], nor for more general states [see point 3) of my previous report].

In summary, I believe that the present manuscript contains sufficient new results to be published in a specialized journal. In particular, with respect to its previous version, motivations and implications are now more clear. However, it does not meet in my opinion the criteria for SciPost Physics.

  • validity: -
  • significance: -
  • originality: -
  • clarity: -
  • formatting: -
  • grammar: -

Author:  John Martin  on 2023-07-05  [id 3781]

(in reply to Report 2 on 2023-05-21)

List of changes: - Added Sec. 7 describing a protocol for accessing Wehrl moments experimentally (see https://arxiv.org/abs/2205.15095v3) - Update of the abstract, introduction, conclusion and references to reflect changes to the main text

Finally, we would like to point out that, compared to the first version of the work we submitted, we have also added discussions on GME prediction (i) based on noisy Wehrl moments and (ii) for mixed states. Each of these discussions indicates that the method of estimating GME with ANNs based on Wehrl moments is a viable method.

Author:  John Martin  on 2023-07-05  [id 3779]

(in reply to Report 2 on 2023-05-21)
Category:
answer to question
reply to objection

In response to the referee's main concern, we now propose an experimental protocol for measuring Wehrl moments. It is based on the concept of spherical t-design and works for any state. Therefore, we have now reintroduced our claim that Wehrl moments are experimentally accessible quantities. To our knowledge, we are the first to propose an experimental protocol for measuring Wehrl moments.

The protocol we propose requires a number of Stern-Gerlach experiments that increases as the square of the number of qubits instead of the cube as for full mixed-state tomography. This type of advantage is considered to be limited by the referee. It should be noted, however, that this same type of advantage is obtained by Grover's algorithm (square root advantage), which nevertheless makes it a reference algorithm. We also trained ANNs on the basis of Wehrl moments obtained with spherical t-designs. Our results confirm that ANNs still predict the GME with high accuracy, even when the condition for the exact determination of the Wehrl moments, i.e. t=Nq, is not met. In addition, it appears that some of the information obtained by a Stern-Gerlach experiment is not used in our specific protocol. This leaves room for improvements in the scalability of the measurement of Wehrl moments.

With regard to point 3) of the referee's previous report, we would like to emphasise that we have indeed considered more general states than only pure states as in the first version of this work. More specifically, we considered mixed depolarised states and showed that our approach still works for this type of state. In passing, we have conceived a numerical approach for computing the GME of these mixed states.

Finally, with regard to point 5) of the referee's previous report, we think that the references they mentions are difficult to compare with our approach, which is based on partial state information in the form of moments of functions in phase space. Furthermore, we are already comparing our ANN-based method with convergence acceleration algorithms.

---

## Round 2 · Author Response

Dear Editor,

We are very appreciative of the comments and questions we got from the referees, as they helped us improve our manuscript significantly. In this revised version, we have provided substantial addaitional scientific works, not simply rewriting. Noteworthy, we have now included a whole new section (Sec. 6) on how to address the estimation of entanglement in mixed states, a new Appendix E presenting how semidefinite programming can be used for calculating the geometric measurement of entanglement of these states (which is not an obvious task), as well as new Appendix D discussing the requested case of noisy Wehrl moments.

Sincerely,
Jérôme Denis, François Damanet and John Martin.

---

## Round 2 · List of Changes

• Update of the title
• Update of the abstract, discussion, conclusion, acknowledgements and references to match with the changes in the main text
• Update of the introduction to match with the changes in the main text and to respond to the comment of the second referee about the measurement of Wehrl moments
• Update of Sec. 4.4 to give an introduction to ANNs and a better explanation of the behaviour of ANNs with respect to qmax and N
• Added Sec. 6 on mixed states
• Added Appendix D on noisy Wehrl moments
• Added Appendix E on the computation of the GME for mixed multiqubit symmetric states

---

## Round 3 · Referee Report · Anonymous (Referee 1) · 2023-7-20

Report

Recently there has been a huge effort to find new ways to learn properties of quantum states without performing full tomography. Moreover, advanced machine learning techniques have been used to handle the large amounts of data generated by experiments with many qubits. Along this direction the authors clearly demonstrate that already a few Wehrl moments can be used to extract relevant information, in this case bounds on the geometric measure, of a state using ANN. They demonstrate that their technique is applicable to mixed states, and they extended it to noisy Wehrl moments, which is of practical relevance. Moreover, in the revised version they devise a method to measure Wehrl moments in experiments based on Stern-Gerlach experiments, which I would assume to be experimentally feasible. Concerning the improvement in scaling with N^2 over N^3 for full tomography I want to add here that for many classical computational problems even the slightest improvement in the exponent makes an arguably huge difference. Altogether, the results are definitely novel, and leave enough room for further investigations. For instance, I believe that the procedure to measure Wehrl moments is not optimal and can still be improved, as the authors also state themselves. Moreover, I wonder if it can be extended to mixed permutation invariant states.

Based on that I still recommend to accept the paper based on the fact that the authors “Open a new pathway in an existing or a new research direction, with clear potential for multipronged follow-up work”.

---

## Round 3 · Referee Report · Anonymous (Referee 2) · 2023-8-10

Report

I would like to provide a brief summary of the evolution of this manuscript up to its current third version. In the initial submission, the authors asserted the experimental accessibility of Wehrl moments ("Wehrl moments are experimentally accessible quantities", refer to Section 1 of the first version). After my substantial critique of this assertion, the second version of the manuscript saw the authors retracting this claim, acknowledging that it "was a bit hasty" (see authors' response to my first report). However, in this latest iteration, they have reintroduced this claim based on their assertion of introducing "a protocol for accessing Wehrl moments experimentally" (see "List of changes" in response to my second report). My present report will focus only on this last point (see my previous reports for further comments).

According to the authors, this protocol is briefly elucidated in Section 7.1 of the revised manuscript. It is based on the evaluation of the Husimi function at a specified number of phase-space points (directions), depending on the degree of the moment to be determined and the system's size.

Unfortunately, the details of this protocol are essentially contained in the very brief passage -- six lines of text -- following the unnumbered equation subsequent to equation (41). The explanation boils down to the following points: (1) In principle, the Husimi function at a particular point can be evaluated by performing a Stern-Gerlach experiment on the multiqubit state after an appropriate rotation; (2) Referring to atomic systems, this task can be achieved following the approach detailed in Ref [42].

Regarding the general case (1), I find some doubts about the effective viability of the protocol. Firstly, the necessary rotation is a highly entangling unitary transformation that transforms a separable state (denoted as |\Omega> in the manuscript) into a multiqubit Dicke state. This process is not straightforward experimentally; for example, in terms of quantum circuits, it will require a relatively long sequence of two-qubit entangling gates. Secondly, the Stern-Gerlach measurement on a multi-qubit state necessitate a setup akin to that employed for full tomography.

As for the reference to atomic systems (2), I note that Ref [42] is a publication from 2000, which has garnered minimal attention within the experimental community to the best of my knowledge.

In essence, the concise portrayal provided by the authors on this critical point fails to persuade me regarding the practicality of their introduced experimental protocol for Husimi function measurements, let alone subsequent Wehrl moment evaluation --- which requires the measurement of the Husimi function at a number of points that increases quadratically with the size of the system and the degree of the moments.

To be clear, let me explicitly mention: I do not have, nor have I ever had, reservations regarding the potential experimental estimation of Wehrl moments "as a matter of principle" (such as through full tomography). Naturally, my apprehensions pertain to their "practical" attainability.

Additionally, I acknowledge the perspective of Referee 1, which highlights the general merit of the manuscript's motivation. I concur with the notion that exploring novel methods for extracting valuable information from high-dimensional systems while bypassing the extensive measurements required for full tomography is worthwhile. The motivations underpinning the manuscript are not areas of contention for me, and have never been.

In conclusion, I am unable to recommend the publication of this manuscript within this journal. The full motivations of my stance are expressed in my previous reports. Regarding this last iteration of the review process in particular, my position rests on the authors' inability to convince me of the experimental viability of their proposed approach.
  • validity: -
  • significance: -
  • originality: -
  • clarity: -
  • formatting: -
  • grammar: -

Author:  John Martin  on 2023-08-24  [id 3924]

(in reply to Report 2 on 2023-08-10)
Category:
reply to objection

Firstly, we would like to thank the referee for his/her time and assessment, which undoubtedly helped us to improve our work. We would like to present a brief summary of how we have responded to all the criticisms made by the referee in his/her previous reports, which ultimately led to the third version of our manuscript. We also wish to express our strong disapproval of the argumentation given by the referee in his/her third report regarding the experimental viability of our approach, which, as we explain below, is clearly based on an erroneous statement.

In his/her first report, the referee raised two major concerns : (1) the lack of evidence that Wehrl moments (WMs) are relatively easy to measure experimentally and (2) the fact that WMs would only be known approximately. He/she also noted ”some other relevant points that the authors should address” : (3) how the ANN would generalize to the mixed states case, (4) the fact that the approach via WMs might not be necessarily easier than full tomography for symmetric multiqubit pure (SMP) states and (5) that we should analyze more in details the connection with previous studies.

In our first revised manuscript, we dealt in depth with points (2), (3) and (5). By introducing Gaussian noise, we showed that training an artificial neural network on noisy Wehrl moments can still provide a good estimate of the geometric measure of entanglement. We also generalized our method for mixed states, showing its effectiveness up to 5% loss of purity. In addition, for certain decoherence channels such as the depolarisation channel, we showed that our method still works even with a high loss of purity. Finally, we have incorporated the references recommended by the referee which we considered to be relevant, and pointed out that comparisons with certain previous works were sometimes difficult due to differences in the physical quantities studied. With regard to points (1) and (4), as we did not have a clear protocol to measure the first Wehrl moments at that stage, we acknowledged that ”our claim that it is possible, in experiments, to access the first WMs for SMP states composed of several qubits was a bit hasty in view of the references we cited”. As a result, in the second version of the manuscript, we have removed certain sentences from the introduction to eliminate any ambiguity. In his/her second report, the referee considered that ”the manuscript still did not meet the SciPost Physics criteria” because the question of the experimental accessibility of the Werhl moments was still open.

We then continued our efforts and finally found a protocol that addresses point (4) raised by the referee, which we present in the 3rd version of the manuscript. However, it seems that the referee is not convinced by our protocol, his/her main criticism being as follows: ”I have some doubts about the actual viability of the protocol. Firstly, the necessary rotation is a highly entangling unitary transformation that transforms a separable state (denoted by |Ω⟩ in the manuscript) into a multiqubit Dicke state. This process is not straightforward experimentally; for example, in terms of quantum circuits, it will require a relatively long sequence of two-qubit entangling gates. Secondly, the Stern- Gerlach measurement on a multi-qubit state necessitate a setup akin to that employed for full tomography”.

Let us now explain that his/her doubts about the viability of our protocol are totally unfounded, because they stem from a fundamental misunderstanding: the unitary transformation that must be applied on a state ρ in our protocol is in fact non-entangling, being only a spin rotation, equivalent to a symmetric local unitary transformation. The reason is that the multiqubit Dicke state |D_N^(0)⟩ is itself separable, unlike the Dicke states |D_N^(k)⟩ with k different from 0 and N. Indeed, we have that |D_N^(0)⟩ = |Ω0⟩ with Ω0 = (0, 0). We recognise that this is not explicitly mentioned in the manuscript and are willing to add it, i.e. make it clear that |D_N^(0)⟩ = |Ω0⟩ with Ω0 = (0, 0) and |Ω⟩ = R(Ω)|Ω0⟩ = R(Ω)|D_N^(0)⟩. In any case, this observation completely invalidates the first part of the referee’s argument. In addition, measuring the Husimi function by determining the probability that a multiqubit state is in different pure separable states using local rotations is a fairly common technique, see e.g. Refs. [1-3]. As stated by G. S. Agarwal in Ref. [1], this is a protocol which is feasible for single spin systems, collections of two-level systems and even for light polarization. In fact, it has already been routinely implemented in several experiments [4-6], e.g. using half-wave plates and polarising beam splitters in the case of multiphoton polarization states [6]. Regarding the second part of the argument, let us stress that the only way to access information about a (multi-qubit) quantum state is to carry out a measurement, and that’s precisely what a Stern-Gerlach experiment can do! This type of measurement is ubiquitous, so it’s not surprising that it also appears in full tomography protocols, which have already been implemented experimentally. The two doubts raised by the referee therefore in no way call into question the possibility of implementing the protocol we are proposing, using the same techniques as in the above-mentioned experiments.

The main advantage of our protocol lies in its N^2 scaling, as opposed to the N^3 scaling associated with full tomography. This point is clearly acknowledged by another referee who writes ”Concerning the improvement in scaling with N^2 over N^3 for full tomography I want to add here that for many classical computational problems even the slightest improvement in the exponent makes an arguably huge difference.” and this in our view convincingly addresses point (4) of the referee. Therefore, we maintain that our Wehrl moment measurement protocol is experimentally viable and offers an advantage over full state tomography, thus addressing both points (1) and (4) of the referee. In addition, we emphasise that our protocol uses only a partial subset of the information that can be obtained with a Stern-Gerlach device. Although our approach may have limitations, it provides a clear pathway for future improvements and prospects for further studies.

The referee also repeatedly criticised the brevity of the presentation of our protocol, describing it as ”very brief”. We are willing to expand this section further and add additional references if deemed necessary. However, we do not consider that an argument of brevity is such as to call into question the scientific validity and relevance of our proposal. In addition, we think it is important to note that our new section on measuring Wehrl moments spans almost two pages and that the central idea is not limited to ”six lines of text following the unnumbered equation subsequent to equation (41)” as stated by the referee, but is based on the key concept of spherical t-design introduced even before Eq. (41).

In conclusion, we are convinced that we have responded to all the relevant and scientifically well-founded criticisms raised by the referee during the review process. This has improved our manuscript considerably and we are grateful to the referee for this. We are ready to add further details to the description of the protocol if this is deemed necessary, and with this we hope that our work can be published in SciPost Physics based on the criteria ”Open a new pathway in an existing or a new research direction, with clear potential for multipronged follow-up work”.

Sincerely yours,
J. Denis, F. Damanet & J. Martin

[1] G. S. Agarwal, Phys. Rev. A 57, 671 (1998).
[2] B. Koczor, R. Zeier and S. J. Glaser, Phys. Rev. A 101, 022318 (2020).
[3] M. A. Perlin, D. Barberena, and A. M. Rey, Phys. Rev. A 104, 062413 (2021).
[4] R. Schmied and P. Treutlein, New J. Phys. 13, 065019 (2011).
[5] T. Chalopin et al., Nat. Comm. 9 4955 (2018).
[6] C. R. Müller et al., New J. Phys. 14, 085002 (2012).

---

## Round 3 · Referee Report · Anonymous (Referee 2) · 2023-10-9

Report

I am now satisfied with the response provided by the authors. It appears that the misunderstanding in the last round of the refereeing process was due to my misinterpretation of the manuscript notation, and for that, I apologize. To enhance clarity, I would suggest that the authors explicitly state in words that the operation involved is non-entangling.

I am now convinced that the protocol introduced in the latest version of the manuscript involves measurements that are not only theoretically feasible but, more importantly, of practical relevance. As a point of fact, this effectively addresses the primary concern I raised in my very first report, particularly regarding the claimed experimental ease of measuring Wehrl Moments (WMs). Specifically, in my first report, I pointed out that I did "not find in the manuscript sufficient support to the claim that WMs are easy to obtain experimentally, which is the practical motivation that underpins this analysis and the "perspectives" that it might open". These concerns are now dissipated.

Therefore, given the latest clarification, the significant changes in the manuscript during the refereeing process, and the substantial progress made in addressing the concerns, I can now recommend the publication of the present manuscript.

---

## Round 3 · Author Response

Dear Editor,

We thank the referees for their reports and comments. In this second revision of our work, we now address the first referee's main concern regarding the experimental implementability of the method and the experimental determination of Wehrl moments. We also respond to all other more minor points.

All our additional work compared with the first version shows that the method of estimating GME using ANNs based on Wehrl moments is a viable method.

Sincerely yours,
J. Denis, F. Damanet, and J. Martin.

---

## Round 3 · List of Changes

• Added Sec. 7 describing a protocol for accessing Wehrl moments experimentally
  • Update of the abstract, introduction, conclusion and references to reflect changes to the main text

Finally, we would like to point out that, compared to the first version of the work we submitted, we have also added discussions on GME prediction (i) based on noisy Wehrl moments and (ii) for mixed states.

---

## Editorial Decision

published